# Viral RNA switch mediates the dynamic control of flavivirus replicase recruitment by genome cyclization

Zhong-Yu Liu[1,2], Xiao-Feng Li[1,2], Tao Jiang[1,2], Yong-Qiang Deng[1,2], Qing Ye[1], Hui Zhao[1], Jiu-Yang Yu[1], Cheng-Feng Qin[1,2]*

[1]Department of Virology, Beijing Institute of Microbiology and Epidemiology, Beijing, China; [2]State Key Laboratory of Pathogen and Biosecurity, Beijing, China

**Abstract** Viral replicase recruitment and long-range RNA interactions are essential for RNA virus replication, yet the mechanism of their interplay remains elusive. Flaviviruses include numerous important human pathogens, e.g., dengue virus (DENV) and Zika virus (ZIKV). Here, we revealed a highly conserved, conformation-tunable *cis*-acting element named 5′-UAR-flanking stem (UFS) in the flavivirus genomic 5′ terminus. We demonstrated that the UFS was critical for efficient NS5 recruitment and viral RNA synthesis in different flaviviruses. Interestingly, stabilization of the DENV UFS impaired both genome cyclization and vRNA replication. Moreover, the UFS unwound in response to genome cyclization, leading to the decreased affinity of NS5 for the viral 5′ end. Thus, we propose that the UFS is switched by genome cyclization to regulate dynamic RdRp binding for vRNA replication. This study demonstrates that the UFS enables communication between flavivirus genome cyclization and RdRp recruitment, highlighting the presence of switch-like mechanisms among RNA viruses.

*For correspondence: qincf@bmi.ac.cn

**Competing interests:** The authors declare that no competing interests exist.

## Introduction

*Cis*-acting regulatory elements play essential yet diverse roles in the life cycle of RNA viruses (*Liu et al., 2009*; *Nicholson and White, 2014*). These elements have been found to regulate viral replicase binding and viral RNA (vRNA) synthesis (*Filomatori et al., 2006*; *Vogt and Andino, 2010*; *Wu et al., 2009*), nucleocapsid assembly (*Goto et al., 2013*; *Keane et al., 2015*; *Morales et al., 2013*), and viral translation (*Brierley and Dos Ramos, 2006*; *Kieft, 2008*; *Liu et al., 2009*; *Simon and Miller, 2013*). In particular, local RNA elements in viral genomes are often involved in long-range RNA-RNA interactions (*Alvarez et al., 2005*; *Nicholson and White, 2014*; *Shetty et al., 2013*), which are often referred to as genome cyclization and can bring regions separated by several thousand nucleotides into proximity. Such strategies have been observed in different classes of RNA viruses and play different roles in virus propagation (*Nicholson and White, 2014*). In fact, the entire genomes of some RNA viruses are highly structured and have complex global architecture (*Athavale et al., 2013*; *Wu et al., 2013*).

The genus flavivirus includes important causative agents for human diseases, such as dengue virus (DENV1-4), West Nile virus (WNV), Japanese encephalitis virus (JEV), yellow fever virus (YFV) and the emerging Zika virus (ZIKV) (*Fauci and Morens, 2016*; *Mlakar et al., 2016*). Major phylogenetic groups (*Blitvich and Firth, 2015*; *Moureau et al., 2015*) of flaviviruses include mosquito-borne flaviviruses (MBFVs), tick-borne flaviviruses (TBFVs), insect-specific flaviviruses (ISFVs) and flaviviruses with no known vectors (NKVs). Leading to diseases with symptoms ranging from mild fever and rash to severe hemorrhagic fever and encephalitis, the emergence and re-emergence of flaviviruses always arouses global concern. The flaviviruses are single-stranded, positive-sense RNA viruses, and

**eLife digest** Flaviviruses include a large family of viruses that are harmful to human health, such as dengue virus, West Nile virus and Zika virus. Understanding the details of the life cycle of these viruses is important for better controlling and treating the diseases that they cause.

The genetic information of flaviviruses is stored in single-stranded molecules of RNA. To form new copies of a virus, the RNA must be replicated in a process that involves two critical steps. First, an enzyme called viral RNA polymerase NS5 must be recruited to a specific end of the RNA strand (known as the 5′ end). Then, the ends of the RNA strand bind together to form a circular loop. However, little is known about whether these two processes are linked, or how they are regulated.

Using bioinformatics, biochemical and reverse genetics approaches, Liu et al. have now identified a new section of RNA in the 5′ end of the flavivirus RNA, named the 5′-UAR-flanking stem (or UFS for short), which is critical for viral replication. The UFS plays an important role in efficiently recruiting the NS5 viral RNA polymerase to the 5′ end of the flavivirus RNA.

After the RNA forms a circle, the UFS unwinds. This makes the NS5 polymerase less likely to bind to the 5′ end of the RNA. Stabilizing the structure of the UFS impairs the ability of the RNA strand to form a circle, and hence reduces the ability of the RNA to replicate. Thus, the UFS links and enables communication between the processes that form the flavivirus RNA into a circle and that recruit the viral RNA polymerase to the RNA.

The structural basis of the interaction between the flavivirus RNA 5′ end, including the UFS element, and the viral RNA polymerase now deserves further investigation. It will be also important to explore whether other types of viruses regulate their replication via a similar mechanism.

their genome is approximately 11 kb and contains a single ORF, which encodes a polyprotein precursor with more than 3000 residues. The precursor is further processed into 3 structural proteins (capsid, pre-membrane/membrane and envelope) and at least 7 nonstructural proteins (NS1, NS2A, NS2B, NS3, NS4A, NS4B and NS5). The largest nonstructural protein, NS5, performs methyltransferase (*Egloff et al., 2002*; *Zhou et al., 2007*), guanylyltransferase (*Issur et al., 2009*) and RNA-dependent RNA polymerase (RdRp) activity (*Nomaguchi et al., 2003*; *Uchil and Satchidanandam, 2003*; *Yap et al., 2007*). NS5 interacts with vRNA, other viral nonstructural proteins and host factors to assemble to the viral replication complex essential for vRNA synthesis (*Klema et al., 2015*). On the other hand, the highly structured 5′ and 3′ untranslated regions (UTRs) occupy the termini of the viral genome (*Friebe and Harris, 2010*; *Gebhard et al., 2011*; *Villordo et al., 2016*) and contain *cis*-acting elements crucial for vRNA replication (*Clyde et al., 2008*; *Filomatori et al., 2006*; *Liu et al., 2013*; *Lodeiro et al., 2009*; *Rouha et al., 2011*; *Villordo et al., 2010*), translation (*Chiu et al., 2005*; *Clyde and Harris, 2006*), viral pathogenesis (*Chapman et al., 2014*; *Funk et al., 2010*; *Roby et al., 2014*) and host adaptation (*Villordo et al., 2015*).

Hybridization of complementary sequences at the 5′ and 3′ termini, which include the 5′-3′ upstream AUG region (UAR) (*Alvarez et al., 2005*; *Zhang et al., 2008*), downstream AUG region (DAR) (*Friebe and Harris, 2010*; *Friebe et al., 2011*) and cyclization sequence (CS) (*Khromykh et al., 2001*) elements, circularizes the genome of MBFV. Genome cyclization, which is also modulated by the downstream of 5′ CS pseudoknot (DCS-PK) in the capsid-coding region (*de Borba et al., 2015*; *Liu et al., 2013*), is crucial for the translocation of NS5 from the 5′ stem-loop A (SLA) promoter to the RNA synthesis initiation site at the 3′ end (*Filomatori et al., 2006*). However, the mechanistic details of this critical process are poorly understood. The 5′ SLA structure and genome cyclization strategy were also identified in other phylogenetic groups of the flavivirus genus, although the specific details differ. The 5′ UAR element in MBFV usually participates in the formation of a local hairpin, stem-loop B (SLB), in the linear conformation of the vRNA. The SLA and SLB elements are separated by a short uracil-rich sequence, which has been shown to mediate vRNA replication of DENV (*Lodeiro et al., 2009*). In the present study, we demonstrated that the U-rich region of MBFV is involved in the formation of a conserved RNA duplex that is designated as a 5′-UAR-flanking stem (UFS) because it locks the 5′ UAR/SLB element between its two strands. A combination of our results suggested that the UFS and its cousin elements regulate the binding of viral RdRp to

vRNA dynamically by switching their conformations in response to the long-range interactions between the viral 5′ and 3′ ends. Furthermore, UFS switching is a general replication strategy in flaviviruses with vertebrate hosts, highlighting their role in the host adaptation and evolution of flaviviruses.

## Results

### The UFS structure is conserved among the flavivirus genus

Although the U-rich region in the 5′ end of the MBFV genome has been shown to enhance vRNA replication (*Friebe and Harris, 2010*; *Lodeiro et al., 2009*), there were controversial opinions regarding the detailed structures downstream of SLA in flaviviruses (*Dong et al., 2008*; *Gebhard et al., 2011*; *Liu et al., 2013*; *Villordo and Gamarnik, 2009*; *Zhang et al., 2008*). To clarify the structural characteristics of the region downstream of SLA, representative sequences derived from various flavivirus species were subjected to the *mfold* RNA folding server (*Zuker, 2003*; *Zuker and Jacobson, 1998*). Because the region of interest is involved in genome cyclization, both the 5′ end sequences and the query sequences composed of the 5′ and 3′ ends were analyzed. *Figure 1A* shows the representative results of the RNA structure prediction. The local RNA structures in the 5′ and 3′ ends of DENV4 vRNA are demonstrated, and elements involved in long-range interactions are highlighted. Structure prediction of the 5′ end local structures showed that the U-rich region of most MBFVs (except the YFV clade) forms a duplex with complementary sequences in or near the viral translational starting region (*Figure 1B*). Because this duplex confines the 5′ UAR/SLB element between its two strands, we designated it as the UFS. Furthermore, the UFS duplex was predicted to unwind due to the formation of the panhandle structure between the 5′ and 3′ ends when long-range interactions between the genome termini were considered (*Figure 1—figure supplement 1* and *Supplementary file 1*).

Because the local folding pattern of the 5′ end and the mode of genome cyclization in the YFV clade are different than those in other MBFVs (*Figure 1B*), the corresponding hairpin including the U-rich region in the YFV clade was recognized as ψUFS. Local structures similar to UFS were also identified in ISFVs (*Figure 1B*) and Modoc virus (MODV) among the NKVs (*Figure 1B*), whereas the non-vectored Rio Bravo virus (RBV, *Figure 1B*) was shown to contain a ψUFS structure. The UFS and ψUFS in NKVs were also involved in genome cyclization, similar to the corresponding structures in MBFVs. In contrast, the UFS-like structures were not predicted to be involved in genome cyclization in ISFVs, suggesting that the UFS in MBFVs and UFS-like structures in ISFVs function differently. Finally, the UFS/ψUFS structure was not identified in TBFVs or in the Yokose virus clade of NKVs; instead, a short hairpin with a large loop occupied the analogous location of the UFS in these viruses (*Supplementary file 2*). Taken together, the above results demonstrated the conservation of UFS elements among the flavivirus genus and suggested that the secondary structures of UFS elements are affected by genome conformation.

### In vitro formation of UFS duplexes in different flaviviruses

To investigate whether the duplex conformation of the UFS can indeed exist locally under in vitro conditions, RNA molecules corresponding to the 5′ ends of DENV1-4, JEV and ZIKV were analyzed by selective 2′-hydroxyl acylation analyzed by primer extension (SHAPE). The SHAPE reactivity data were annotated on RNA structure models generated by *mfold* prediction and sequence comparison (*Figure 2*). The SHAPE results were shown to agree well with the predicted RNA structures. It was shown that the UFS element of JEV forms a conical duplex structure under in vitro conditions (*Figure 2A*) because all nucleotides of the UFS exhibited low SHAPE reactivity. Similar results were obtained for DENV serotypes 1, 3, 4 and ZIKV (*Figure 2B,D,E and F*), except that several nucleotides in the ZIKV and DENV1/3 UFS regions had elevated SHAPE reactivity, which was likely due to the presence of internal loops. In contrast, the SHAPE results indicated that the UFS duplex of DENV2 is highly unstable because the DENV2 UFS region showed considerable SHAPE reactivity (*Figure 2C*). This result was consistent with thermodynamic calculations and suggested that the UFS duplex is a highly dynamic structure, at least in some flaviviruses. The presence of the DENV UFS duplex conformation was further confirmed by SHAPE analysis of DENV3 5′ end RNA molecules containing mutations in the UFS (*Figure 2—figure supplement 1*). It was shown that the SHAPE

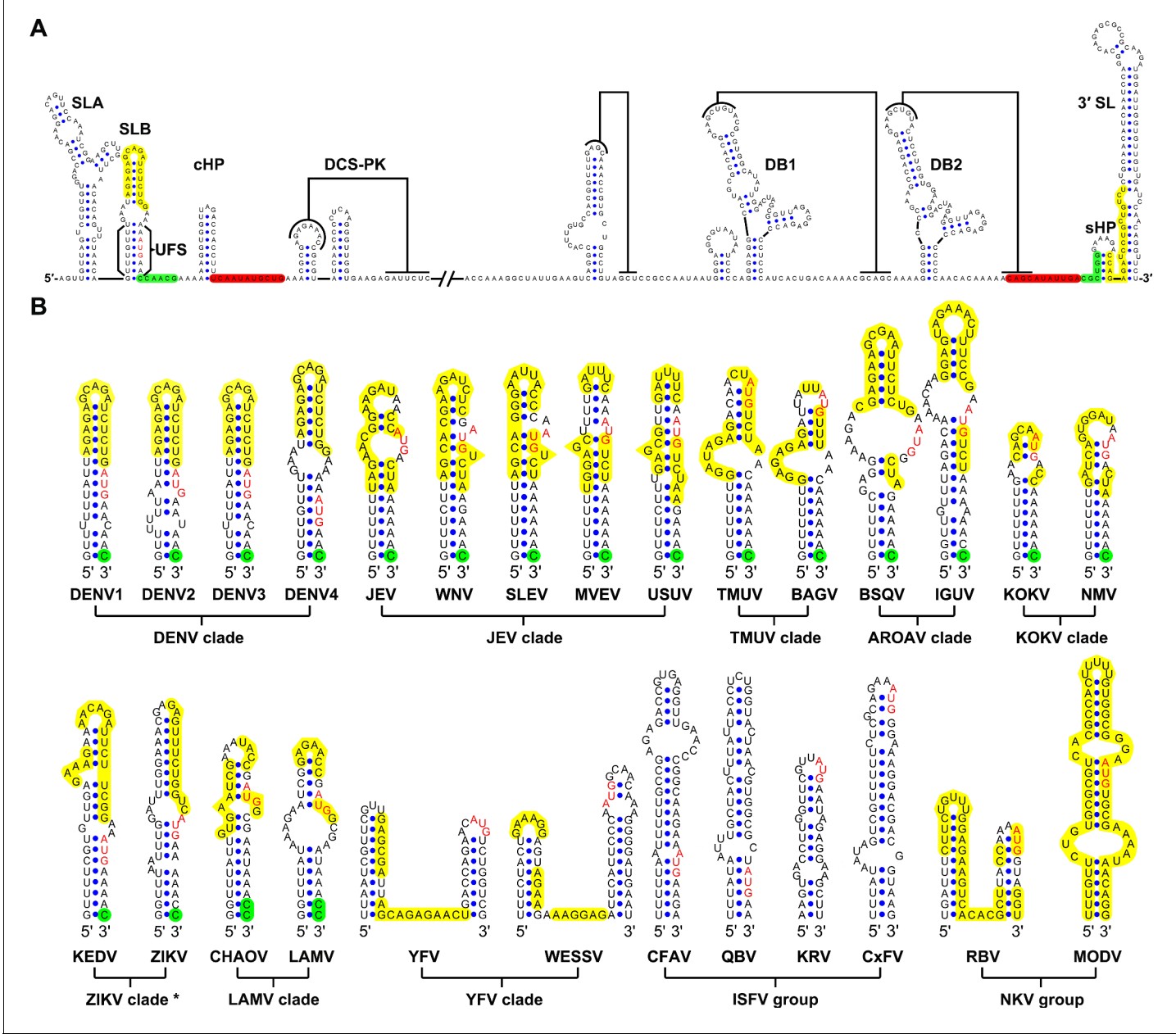

**Figure 1.** Identification and comparison of the 5′ UAR-UFS elements among flaviviruses. (**A**) Terminal RNA structures of the DENV4 genome. The UAR, DAR and CS elements are highlighted in yellow, green and red, respectively. Pseudoknotted interactions are labeled. The UFS stem region is indicated by parentheses. (**B**) Comparison of the 5′ UAR-UFS and UFS-like elements in MBFVs, NKVs and ISFVs. *: KEDV and ZIKV were supposed to belong to the same clade herein. The MBFV 5′ UAR sequences and NKV sequences involved in genome cyclization are colored in yellow. The nucleotides that participate in the DAR interactions of MBFVs are labeled in green. Translational start codons in (**A**) and (**B**) are shown in red. Virus name abbreviations are annotated in *Figure 1—figure supplement 2*.

The following figure supplements are available for figure 1:

**Figure supplement 1.** Circular conformation of the DENV4 genomic RNA.

**Figure supplement 2.** Molecular phylogenetic analysis of the flavivirus genus.

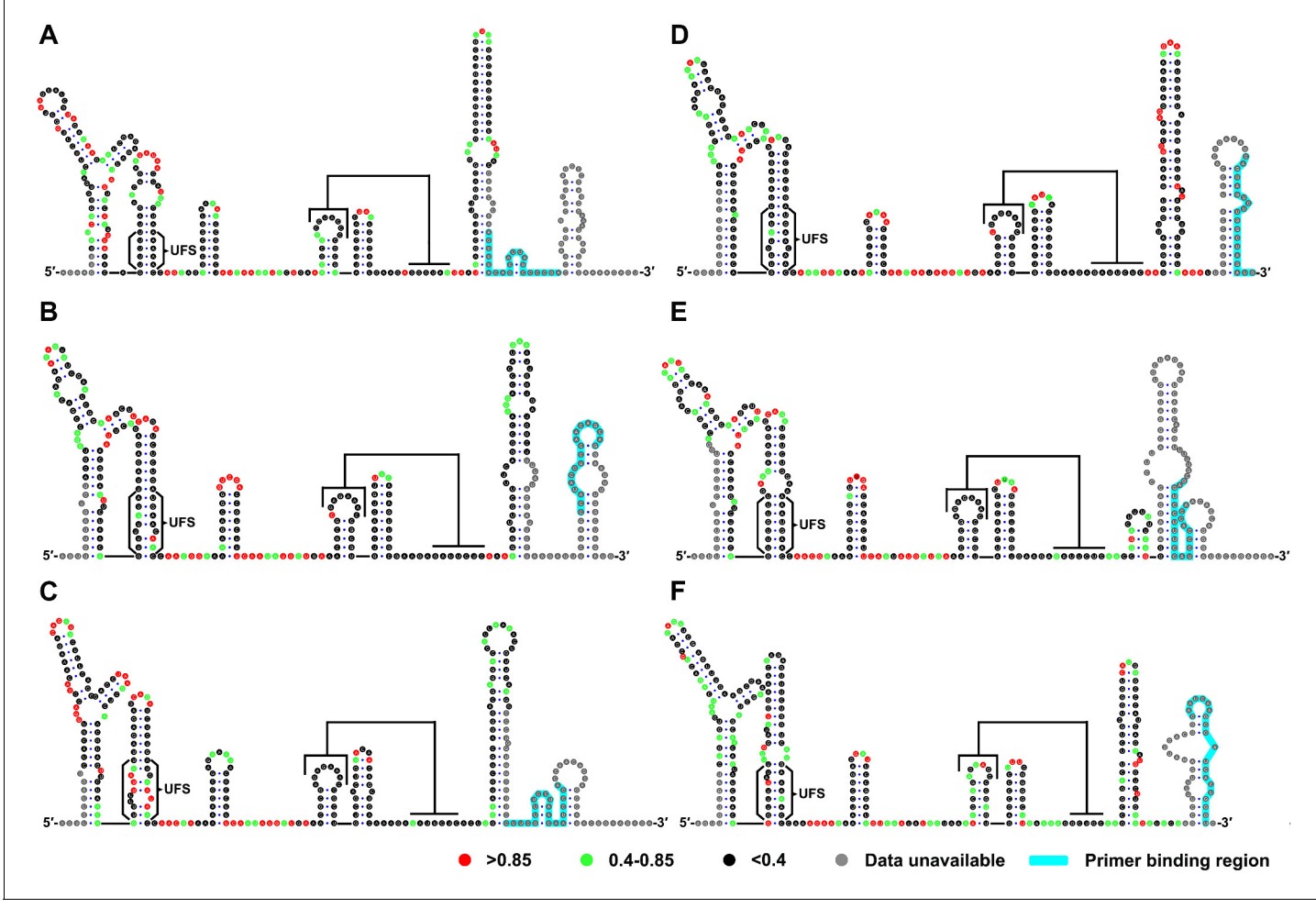

**Figure 2.** SHAPE analysis of the 5′ end RNA of representative flaviviruses. The SHAPE reactivity results are labeled in the structure models of different flaviviruses. Highly reactive nucleotides (SHAPE reactivity > 0.85) are labeled in red, whereas nucleotides with moderate SHAPE reactivity (0.4–0.85) are labeled in green. Black-colored nucleotides indicate low or no SHAPE reactivity (<0.4). Nucleotides lacking SHAPE data are labeled in gray. The primer-binding region in the reverse transcription reaction is shown in light blue. (A) JEV, strain SA-14-14-2 (AF315119). (B) DENV1; the sequence is based on strain WestPac (U88535). (C) DENV2; the sequence is based on strain NGC (KM204118). (D) DENV3; the sequence is based on strain 80–2 (AF317645). (E) DENV4; strain 814669 (AF326573), with two artificial nonsense mutations that do not affect the 5′ end secondary structures or vRNA replication. The two mutations, located in the loop region of cHP and DCS-PK Loop 3, respectively, are labeled in black-colored font. (F) ZIKV; strain FSS13025 (KU955593). Results presented were from two biological replicates.

The following source data and figure supplement are available for figure 2:

**Source data 1.** Source data for *Figure 2*.

**Figure supplement 1.** SHAPE analysis of the DENV3 UFS mutants.

reactivity of the UFS region significantly increased in UFS-disrupted mutants (*Figure 2—figure supplement 1*, D3-M13A and D3-M13B), and restoration of UFS base-pairing endowed the corresponding region with low SHAPE reactivity (*Figure 2—figure supplement 1*, D3-M13C). The above results demonstrated that the UFS duplex can exist locally in the vRNA 5′ end under in vitro conditions, suggesting that the UFS can assume the duplex conformation, at least under certain states of the flavivirus genome.

## The UFS duplex is critical for efficient vRNA replication

The UFS is located just downstream of the SLA promoter element, and it interlocks with cyclization sequences in the vRNA 5′ end. This peculiar localization of the UFS suggests a unique role in vRNA replication. Functional mutagenesis of the UFS was performed using a DENV4 replicon with internal ribosome entry site (IRES)-controlled viral translation (*Liu et al., 2013*). To eliminate the possibility that productive 5′-cap-directed translation is caused by mutations altering the start codon (*Clyde and Harris, 2006*), which is embedded in the DENV4 UFS duplex, a nonsense mutation was introduced into the variable loop 3 of DCS-PK in the background of p4-cHPstop-SP-IRES-Rluc-Rep to generate the p4-Dualstop-SP-IRES-Rluc-Rep replicon (*Figure 3A*). The replication characteristics of the two replicons were basically the same (*Figure 3B*). First, a panel of mutants was generated, and their replication efficiency was assessed in BHK-21 cells. Disrupting the base-pairing of the UFS

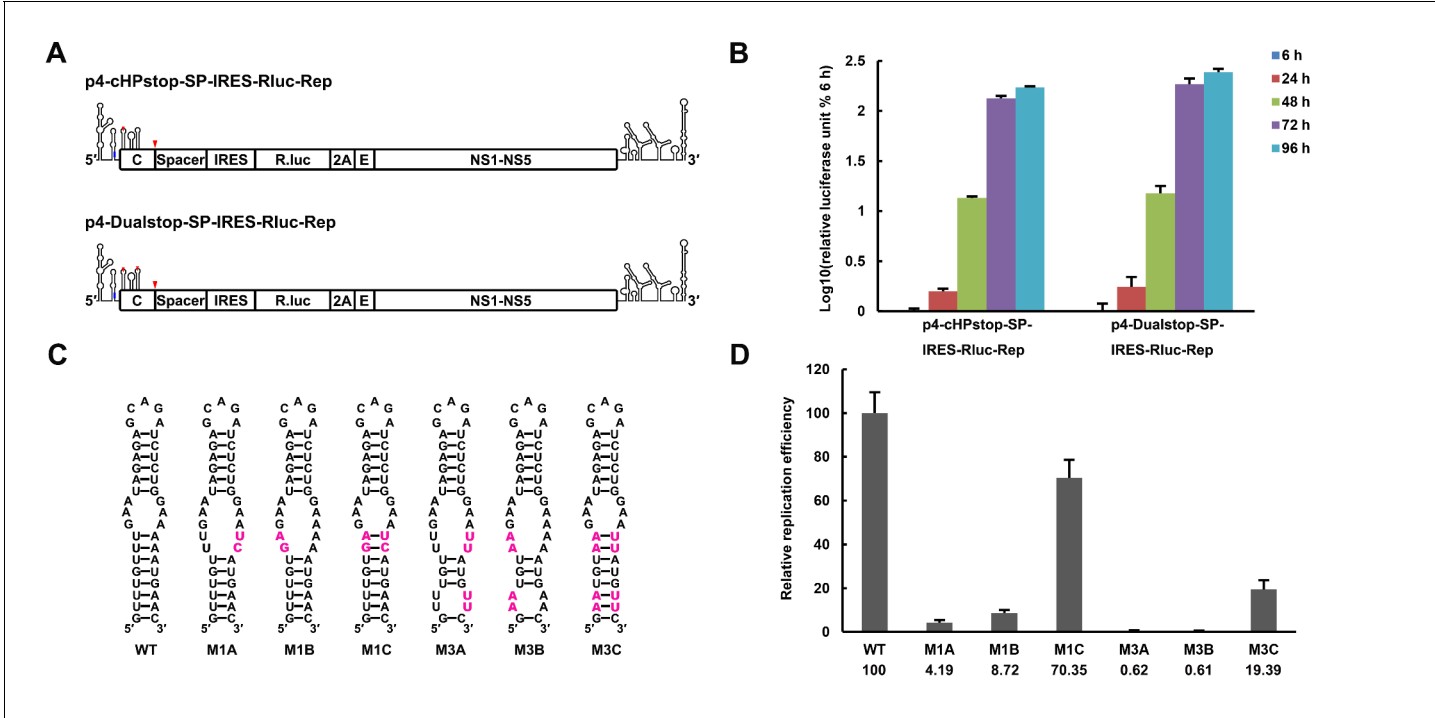

**Figure 3.** The secondary structure of the UFS is required for flavivirus vRNA replication. (**A**) The organization of the DENV4 replicon constructs. In p4-cHPstop-SP-IRES-Rluc-Rep, the translation of viral nonstructural proteins is controlled by the EMCV IRES, and artificial stop codons (red inverted triangles) were introduced into the cHP loop region as well as the end of the capsid-coding region to abolish translation directed by the 5′ cap structure. In the improved version, p4-Dualstop-SP-IRES-Rluc-Rep, one additional stop codon was introduced into loop 3 of the DCS-PK element to abolish the potential translational start from a cryptic AUG in the 5′ CS element. (**B**) Replication characteristics of the above two replicons in BHK-21 cells. Relative luciferase units are defined as the ratio of luciferase units measured at different time points after transfection to the value measured at 6 hr post-transfection. (**C**) Demonstration of DENV4 UFS mutants. Mutations are shown in purple. The 'A' mutants contain mutations in the downstream strand of the UFS, and the 'B' mutants contain mutations in the upstream strand of the UFS. The 'C' mutant in each group combines the 'A' and 'B' mutations to restore the secondary structure of the UFS. Note that the secondary structures shown here are for illustration only and do not represent the actual folding of these mutants. (**D**) Replication efficiency of the UFS mutants. The results of one of two independent biological replicates were shown. One hundred nanograms per well of replicon RNA species were transfected into BHK-21 cells in triplicate. The results are expressed as the percentage of the relative luciferase units of a UFS mutant at 72 hr post-transfection to the value of the WT replicon (relative replication efficiency). The error bar represents the standard deviation in all figures. The mean values of relative replication efficiency are listed below the names of the replicon constructs.

The following source data and figure supplement are available for figure 3:

**Source data 1.** Source data for 3B, 3D and *Figure 3—figure supplement 1*.

**Figure supplement 1.** Replication of DENV4 UFS mutants targeting the AUG region.

(*Figure 3C,D*. M1A, M1B, M3A and M3B) greatly reduced vRNA replication, whereas reconstituting base-pairing by combining the corresponding mutations was able to restore vRNA replication (*Figure 3C,D*. M1C and M3C), although the M3C mutant replicated moderately less efficiently than the wild-type (WT) replicon, possibly due to the apparent primary sequence changes in the U-rich region. Next, the impact of the M1 and M3 mutations on the circularized vRNA structure was assessed using the *mfold* online server. It was shown that the overall genome cyclization pattern was only slightly affected by the M1 and M3 mutations (*Supplementary file 3*), and the changes in the stability of the predicted circularized structures was considerable smaller than the changes in the local UFS structure. To further confirm the function of local UFS structure in vRNA replication, we designed a panel of mutations (*Figure 3—figure supplement 1*), which nether affected vRNA cyclization pattern nor the internal loop structure in the circular form, and the corresponding mutants were assessed for replication efficiency. It was shown that disruption of UFS base pairing (mutant AGU and GAU) reduced vRNA replication greatly, whereas substituting of a few base pairs in the UFS (mutant AUA and ACG) only have moderate effects on vRNA replication.

To confirm that the RNA conformation was changed as expected by the corresponding mutations, SHAPE analysis was performed for DENV4 5′-300 nt RNA containing the M1A, M1C, M3A or M3C mutation, and the SHAPE results were compared with WT RNA (*Figure 4*). In WT RNA, nucleotides composing the UFS only showed low or background SHAPE reactivity, whereas the SHAPE reactivity of the UFS region increased significantly in the M1A and M3A mutants, suggesting that the UFS duplex was indeed disrupted by the corresponding mutations. In contrast, the M1C and M3C mutations greatly reduced the SHAPE reactivity of the UFS region. RNA structure predictions using the *RNAstructure* (*Reuter and Mathews, 2010*) software with SHAPE constraints confirmed that the UFS structure was disrupted/destabilized or reconstituted as aimed, and these mutations did not affect the overall secondary structure of the 5′ end RNA (*Supplementary file 4*). Taken together, the above results demonstrated that the presence of the UFS greatly enhances vRNA replication, and this function is directly correlated to its duplex conformation.

## The function of the UFS is crucial for viral propagation of flavivirus

We further attempted to investigate the influence of the UFS on the propagation of infectious virus. Various mutations targeting the UFS element (*Figures 3* and *5A*) were introduced into a DENV4 infectious clone. vRNA copies in BHK-21 cells were determined at different time points post-electroporation (*Figure 5B–D*). It was shown that the vRNA copies in BHK-21 cells transfected with the M1A, M3A or M5A mutants were significantly lower than those transfected with WT vRNA at 48–72 hr post-transfection, whereas restoration of the UFS element by complementary mutations partially (*Figure 5C*, M3C) or fully (*Figure 5B,D*. M1C and M5C) recovered the vRNA replication levels, highlighting the importance of the UFS for vRNA replication of infection-competent viruses. An indirect immunofluorescence assay (IFA, *Figure 5E*) revealed that the DENV-positive rates were apparently lower in cells transfected with UFS-disrupted mutants than in those transfected with UFS-restored mutants or WT vRNA at the same time points. The virus titers in the culture supernatants of M1A-transfected BHK-21 cells were also significantly lower than those of M1C- or WT-transfected cells (*Figure 5F*).

The effect of UFS mutations on vRNA stability was then assessed. To this end, DENV4 NS5-GVD-vRNAs containing different UFS mutations were transfected into BHK-21 cells by electroporation, and vRNA copies were determined at 4, 24, 48 and 72 hr post-transfection. Only negligible differences in RNA stability were found between WT RNA and the mutants of the M1 or M3 groups (*Figure 5G,H*), and the stability of the M5C mutant was moderately reduced compared with that of the M5A mutant (*Figure 5I*). Thus, the differences in the RNA stability of the UFS mutants did not account for their different replication characteristics. The above results demonstrated that the UFS element is required for virus propagation due to its function in vRNA replication. In agreement with the function of DENV4 UFS, we also showed that disrupting of the UFS greatly reduced viral propagation of ZIKV, whereas reconstitution of the UFS duplex restores ZIKV propagation by IFA tests performed in transfected BHK-21 cells (*Figure 5—figure supplement 1*).

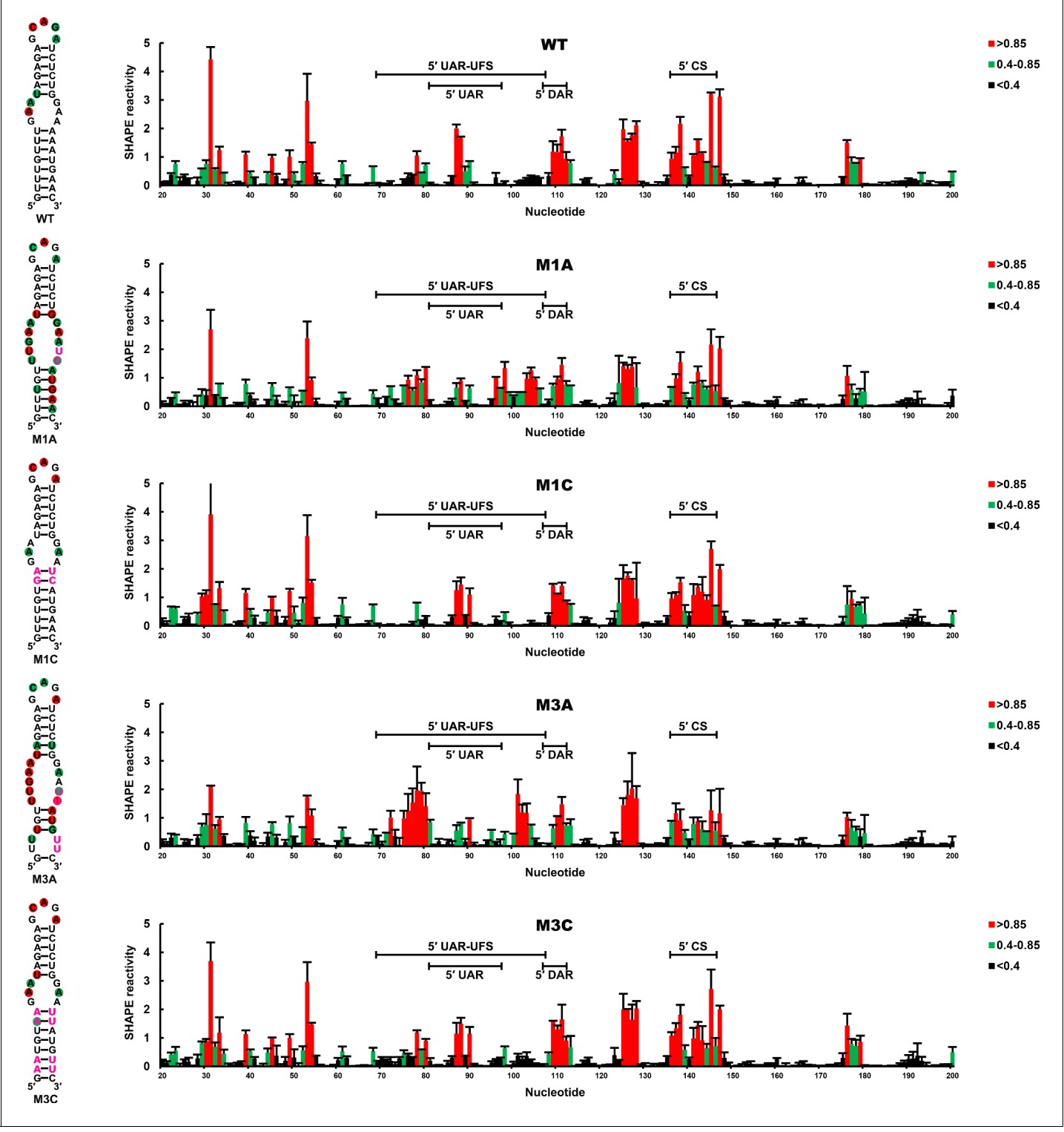

**Figure 4.** SHAPE analysis of the DENV4 UFS mutants. SHAPE analysis was performed for DENV4 5'-300 nt RNA containing the UFS mutations M1A, M1C, M3A or M3C. The SHAPE result for WT 5' end RNA is shown in parallel. SHAPE diagrams of 5' 20–200 nt are shown, and the regions corresponding to various 5' elements are indicated. Negative SHAPE reactivity is set to zero. The SHAPE reactivity of the 5' UAR-UFS region is annotated on the corresponding secondary structures (left). Nucleotides with SHAPE reactivity greater than 0.85 are labeled in red, and moderately reactive sites (0.4–0.85) are labeled in green. Unlabeled sites have low or no SHAPE reactivity (<0.4). Two biological replicates were performed for each RNA, and the error bars represent the standard deviation.

*Figure 4 continued on next page*

*Figure 4 continued*

The following source data is available for figure 4:

**Source data 1.** Source data for *Figure 4*.

## The duplex conformation of the UFS is required for robust NS5Pol binding and RdRp activity

To understand the mechanism of how the UFS fulfills its role in vRNA replication, a series of in vitro assays were performed (*Figure 6*). Because NS5 encodes the RdRp activity required for vRNA synthesis and has specific interactions with the 5′ end of vRNA (*Dong et al., 2008*; *Filomatori et al., 2006*; *Lodeiro et al., 2009*), whether the UFS is involved in the recruitment of NS5 was examined. WT and UFS-mutated DENV4 5′-300 nt RNA molecules were incubated with different concentrations of the NS5 RdRp domain (NS5Pol), and the reaction mixtures were analyzed by native PAGE. The binding of NS5Pol to 5′-300 nt RNA containing the M1A or M1B mutation in UFS was much weaker than the binding to 5′-300 nt WT RNA, whereas the 5′-300 nt M1C and WT RNA bound to NS5Pol with comparable affinity (*Figure 6C*, top and middle panels). The 5′-300 nt M3A and M3B RNA also interacted with NS5Pol weakly, whereas the combination of the two mutations apparently restored the interaction between NS5Pol and 5′-300 nt M3C RNA (*Figure 6C*, bottom panel). The above results indicated that the UFS duplex is required for efficient NS5Pol binding in DENV4. Moreover, EMSA assay demonstrated that the UFS duplex is also important for the efficient binding of JEV NS5Pol to its 5′ end RNA (*Figure 6—figure supplement 1*), suggesting that the function of UFS in viral RdRp recruitment is conserved among flaviviruses.

Next, in vitro RdRp assays using DENV4 5′-160 nt RNA as a template were performed to investigate whether the role of the UFS in NS5Pol recruitment is required for de novo RNA synthesis (*Figure 6D*). Apparently fewer products were generated in the reactions containing templates with disrupted UFS elements compared with the reaction containing the 5′-160 nt WT template, and restoration of the UFS duplex greatly increased the amounts of RdRp products in reactions using the corresponding 5′-160 nt mutants as templates. Moreover, an electrophoretic mobility shift assay (EMSA) experiment using 5′-160 nt RNA as a probe confirmed the function of UFS in NS5Pol recruitment, although the overall affinity of 5′-160 nt RNA for NS5Pol was lower than that of 5′-300 nt RNA (*Figure 6E*). Taken together, the above results demonstrated that the duplex conformation of the UFS element is required for efficient NS5 recruitment and de novo initiation efficiency, which is closely correlated with its function in vRNA replication.

## The stability of the UFS duplex determines its function in vRNA replication

The UFS elements were shown to be rich in UA/AU base pairs, which contribute less to the stability of RNA structures than CG/GC base pairs. Furthermore, internal loops and/or wobble base pairs are prevalent inside the UFS or at the junction connecting the UFS and SLB, which further decreases their structural stability. To investigate whether the low stability of the UFS duplex is associated with its function, UA base pairs of the DENV4 UFS were progressively substituted with GC base pairs, and the replication efficiency of the corresponding mutants was assessed using the above replicon system in BHK-21 cells (*Figure 7*). As expected, all mutants containing disrupted UFS elements showed only extremely low levels of replication (*Figure 7*, M2A, M2B, M4A and M4B). One single UA-to-GC substitution at different positions (*Figure 7*, M2C1 and M2C2) had little effect on vRNA replication, in agreement with the above results (*Figure 3*, M1C). Surprisingly, all UFS mutants containing 2 or more UA-to-GC base pair substitutions replicated poorly compared with the WT replicon (*Figure 7*, M2C and M4C). In contrast, the M5 mutant series, in which only one UA-to-GC substitution was involved, exhibited replication characteristics similar to the M1 and M3 series. The above results indicated that increasing the stability of the UFS duplex is harmful for vRNA replication.

To further confirm this finding and rule out the possibility that the base pair composition itself affects UFS function, another panel of replicon mutants, which targeted the 3-nt internal loop region between the DENV4 UFS and SLB, was generated. Shortening the length of the internal loop to 1 nt

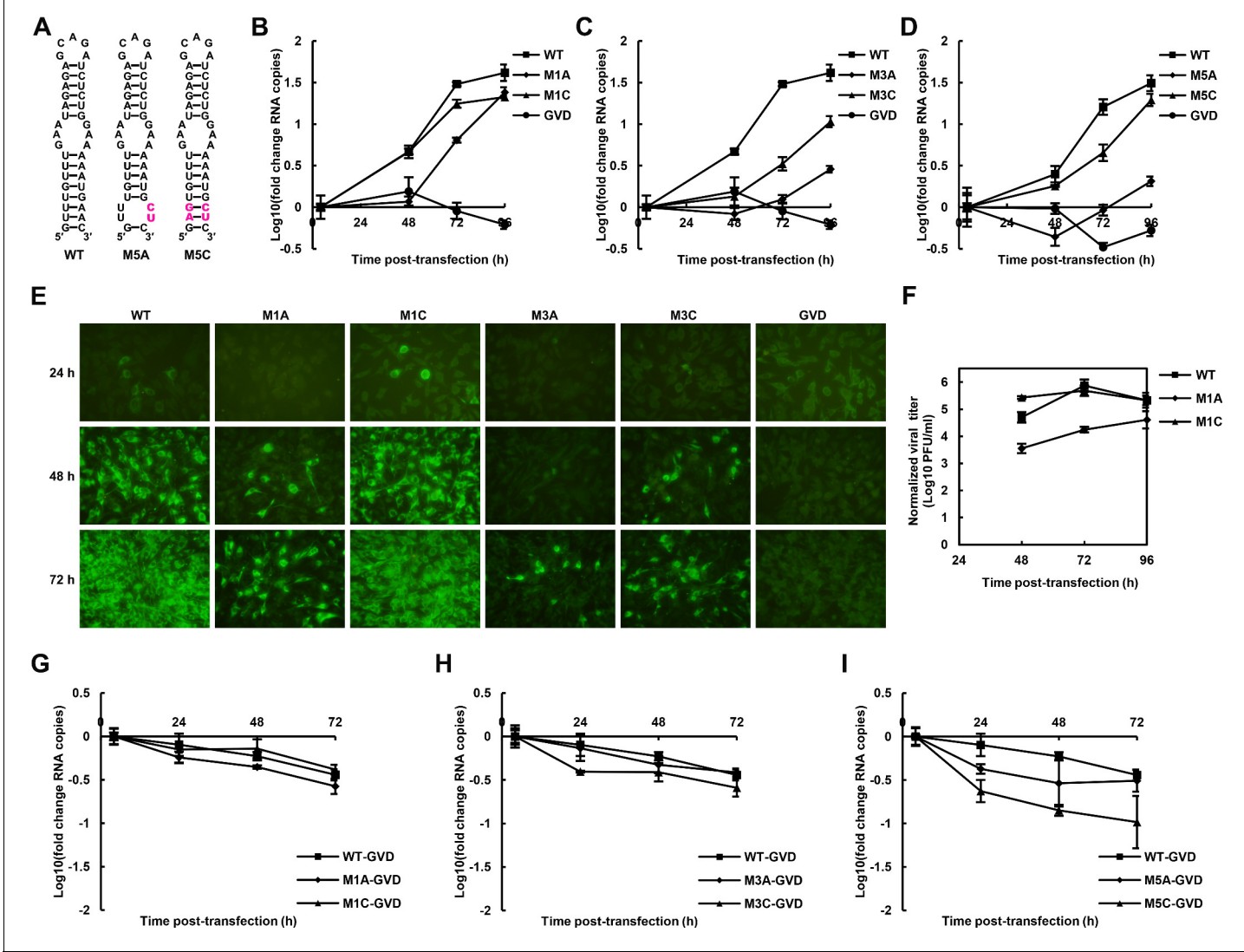

**Figure 5.** The role of the UFS in viral propagation. (**A**) DENV4 UFS M5 series mutants. Mutations are shown in purple. (**B–D**) vRNA replication profiles of different UFS mutants in transfected BHK-21 cells. WT and NS5-inactive GVD mutant vRNA, measured in parallel, are shown as controls. (**B**) M1 series, (**C**) M3 series and (**D**) M5 series. (**E**) Indirect immunofluorescence assay of WT, GVD, and UFS-mutated vRNA-transfected BHK-21 cells. (**F**) Determination of the virus titers in the supernatants of the M1 series mutant-transfected BHK-21 cells. The virus titers in the supernatants of WT vRNA-transfected cells were determined in parallel. The virus titers were normalized according to input vRNA copies determined at 4 hr post-electroporation. (**G–I**) The RNA stability of the UFS mutants was determined by qRT-PCR and are expressed as the fold change relative to the vRNA copies at 4 hr post-transfection. (**G**) M1 series, (**H**) M3 series and (**I**) M5 series. The results were from one biological replicate. Experiments to determine vRNA copies and virus titers were performed in triplicate. Note that the time point '72 hr' for the GVD group in (**D**) contains data from only two parallel wells (technical replicates), because the result of the third sample was unreliable due to an error in the RNA isolation process.

The following source data and figure supplement are available for figure 5:

**Source data 1.** Source data for 5B and 5C.
**Source data 2.** Source data for 5D
**Source data 3.** Source data for 5F
**Source data 4.** Source data for 5G, 5H and 5I.
**Figure supplement 1.** Replication of ZIKV UFS mutants in BHK-21 cells.

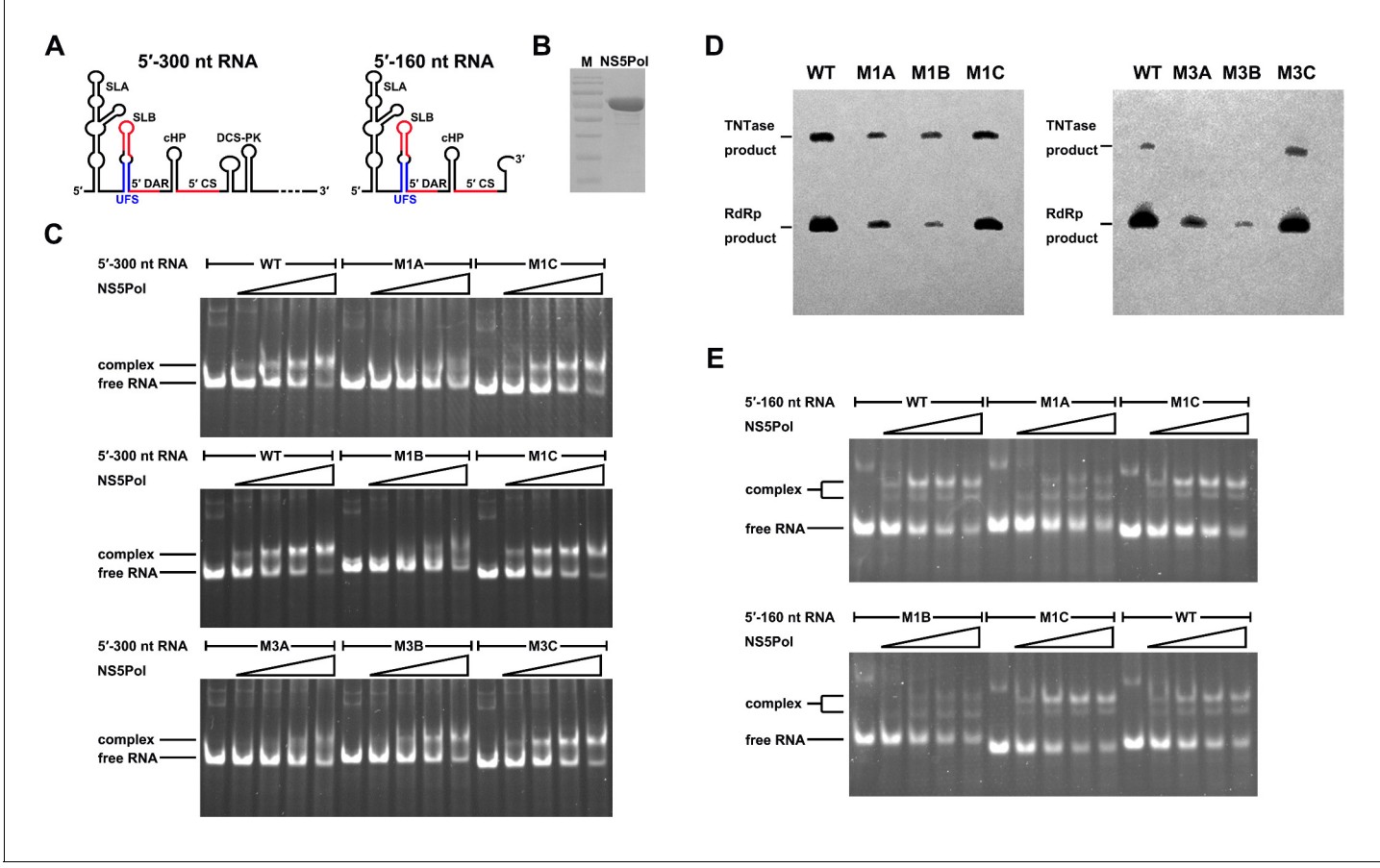

**Figure 6.** UFS is crucial for RdRp recruitment and de novo RNA synthesis. (**A**) Simplified diagrams of DENV4 virus 5′ end RNA constructs used for RdRp binding and/or in vitro RdRp activity assays. The region corresponding to the UFS element is shown in blue, and the red regions represent the genome cyclization elements. (**B**) SDS-PAGE of purified recombinant NS5Pol of DENV4. (**C**) The binding of DENV4 NS5Pol to 5′-300 nt RNA molecules containing UFS mutations was analyzed by EMSA. No NS5Pol was present in the left first lane of each group. The NS5Pol concentrations in the reactions were approximately 2.0, 3.5, 5.0 and 7.5 µM (from left to right). (**D**) In vitro RdRp activity assay using 5′-160 nt RNA as a template. The reactions were incubated at 30°C for 20 min. Detection was based on the incorporation of biotin-11-CTP into the products. Left, the results of the M1 series; right, the results of the M3 series. Two bands were detectable in the blot: the upper template-sized band was generated by the terminal addition of biotin-11-CTP to the input template due to the terminal nucleotide transferase activity of NS5Pol in the presence of $Mn^{2+}$, and the lower band represented the dsRNA product of de novo RNA synthesis, which was confirmed by comparison of the electrophoretic patterns of the RdRp products and annealed dsRNA molecules (*Figure 6—figure supplement 2*). (**E**) EMSA was performed for the 5′-160 nt RNA molecules containing M1 series mutations. The NS5Pol concentrations in the reactions were approximately 3.6, 7.2, 10.7 and 14.3 µM (left to right).

The following figure supplements are available for figure 6:

**Figure supplement 1.** The UFS is required for efficient NS5 binding to 5′ RNA in JEV.

**Figure supplement 2.** Identification of the products of the RdRp reactions.

or 2 nt did not affect vRNA replication greatly. In contrast, total deletion of the internal loop, which caused the SLB to stack onto the UFS, significantly reduced the vRNA replication efficiency (*Figure 7* and *Figure 7—figure supplement 1*, L1D, L2D and L3D). Moreover, replication assays performed for various purine-to-purine and purine-to-pyrimidine substituting mutations targeting the internal loop (*Figure 7—figure supplement 1*) suggested that the purine-rich properties of the internal loop are optimal for efficient DENV4 vRNA replication. Taken together, the above results demonstrated that the stability of the UFS duplex is a critical determinant for its function in vRNA replication. The defects in the vRNA replication of the over-stabilized UFS mutants were not caused by alternations

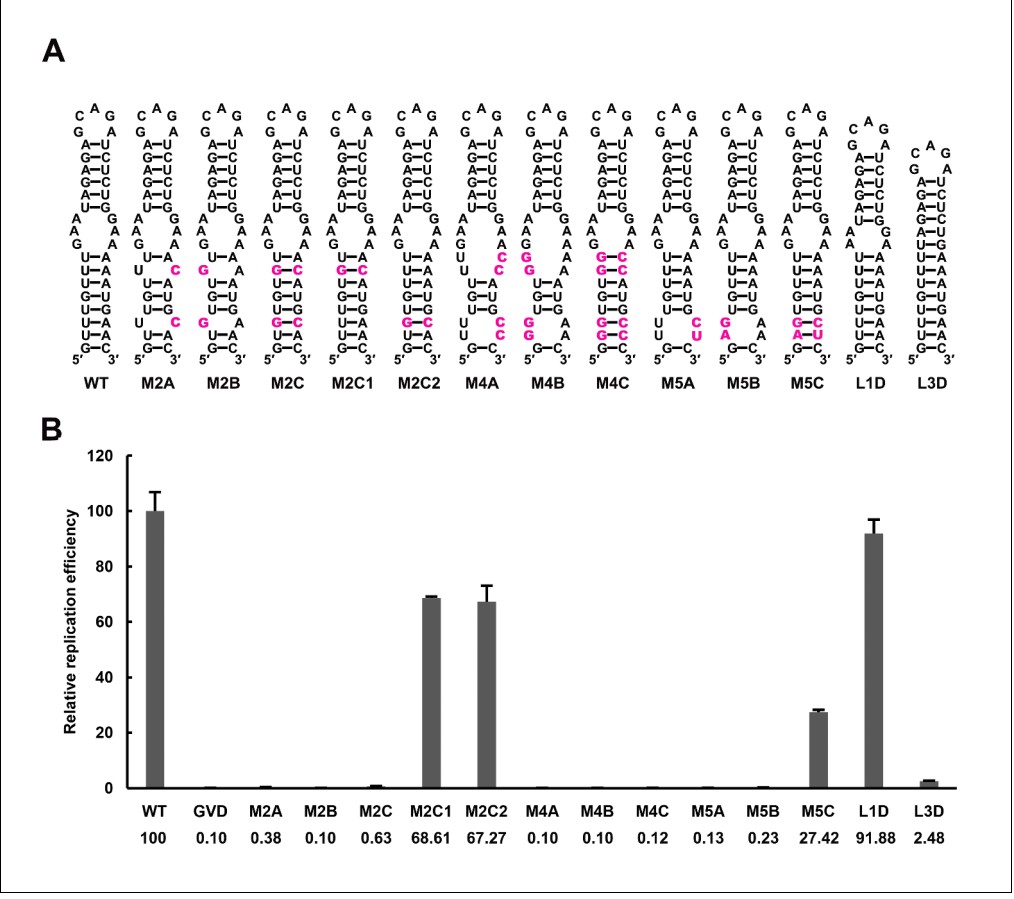

**Figure 7.** Effect of UFS stability on vRNA replication. (**A**) Demonstration of UFS mutants. The mutations are indicated in purple. The M2C mutant contained two UA-to-GC base pair substitutions, whereas the M4C mutant contained four substitutions in the UFS secondary structure. It should be noted that the secondary structures are meant for illustration purposes only, and the structures displayed for UFS-disrupted mutants are not the most thermodynamically favorable ones. (**B**) Relative replication efficiency of UFS mutants shown in (**A**) at 72 hr post-transfection. The results of one of two independent biological replicates were shown. Fifty nanograms per well of replicon RNA were transfected into 96-well plates. Experiments were performed in triplicate.

The following source data and figure supplements are available for figure 7:

**Source data 1.** Source data for *Figure 7* and *Figure 7—figure supplement 1*.

**Figure supplement 1.** Effects of mutations in the SLB-UFS internal loop on vRNA replication.

**Figure supplement 2.** NS5Pol binding assay of the DENV4 5′-300 nt M2A and M2C mutants.

in their affinity for NS5Pol, as an EMSA indicated that the 5′-300 nt M2C RNA bound to NS5Pol efficiently (*Figure 7—figure supplement 2*).

## Over-stabilized UFS hinders genome cyclization

RNA structure prediction (*Figure 1—figure supplement 1* and *Supplementary file 1*) and a previous report (*Sztuba-Solinska et al., 2013*) suggested that the UFS duplex is melted after genome cyclization, whereas the 5′ UAR/SLB element, which is locked by the UFS duplex, is important for flavivirus genome cyclization (*Alvarez et al., 2005*; *Zhang et al., 2008*). We speculated that an over-stabilized UFS duplex would be too difficult to unwind and would hinder 5′-3′ UAR hybridization. To confirm this probability, a DENV4 5′-3′ RNA binding assay was employed (*Figure 8A*). The M1A,

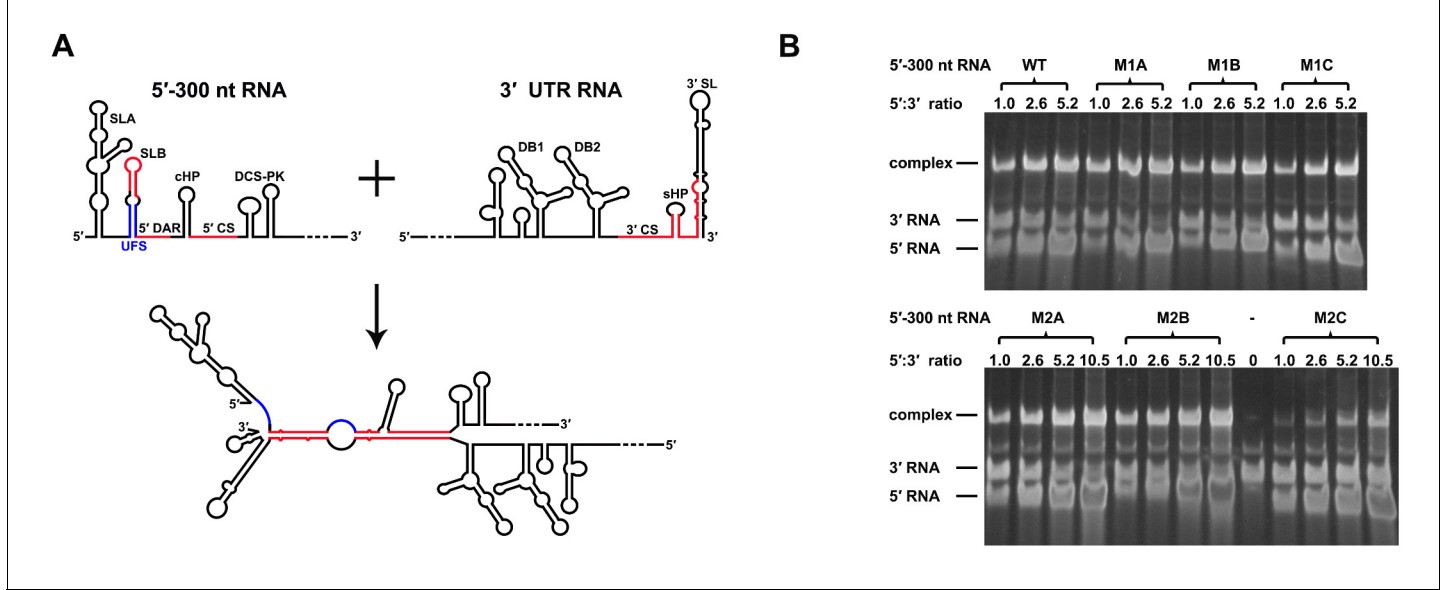

**Figure 8.** Increasing the stability of the UFS hinders vRNA cyclization. (**A**) Schematic diagram of experimental design. As 5′-300 nt RNA is incubated with 3′ UTR RNA, a 5′-3′ RNA bimolecular complex is formed due to the interactions that promote genome cyclization. Various RNA structural elements in the DENV4 genomic ends are shown. The regions involved in terminal interactions are colored in red, and the UFS element is labeled in blue. (**B**) RNA binding assay of the UFS mutants. 3′ UTR RNA (20 ng/µl) was incubated with different amounts (16, 40, 80 and 160 ng/µl) of various 5′-300 nt RNA mutants in 20 µl reactions. The formation of RNA complexes was then analyzed by native TBE PAGE gels. A gel image of the M1 series mutants is shown on the top, and the results for the M2 series mutants are shown below. The molar ratios between 5′ and 3′ RNA are indicated above the gel lanes.

The following figure supplements are available for figure 8:

**Figure supplement 1.** RNA binding assay of the UFS L1D, L3D and M3 series mutants.

**Figure supplement 2.** The calculation of binding affinity of 3′ UTR for the 5′-300 nt RNAs.

M1B and M1C mutations (*Figure 8B*), as well as the M3 series mutations (*Figure 8—figure supplement 1*), had little effect on the formation of 5′-3′ complex. However, when the M2 series mutants were assayed, although the M2A and M2B mutations also did not influence the formation of 5′-3′ RNA complex apparently, the complex formed by the 3′ UTR and 5′-300 nt M2C RNA was greatly reduced compared to the amounts of complexes formed by the 3′ UTR and other 5′-300 nt RNA species (*Figure 8B*). Estimation of dissociation constant (*Figure 8—figure supplement 2*) revealed that the $K_d$ of 5′-300 nt M2C binding with 3′ UTR is approximately ten fold higher than the $K_d$s of the other 5′ RNAs. These results together indicated that increasing the stability of the UFS hinders vRNA terminus interactions. Furthermore, the 5′-300 nt L3D mutant, in which the internal loop was deleted, also interacted with the 3′ UTR poorly (*Figure 8—figure supplement 1*).

SHAPE analysis was then performed to observe the structural changes to 5′ end RNA caused by different concentrations of 3′ UTR RNA (*Figure 9*). When the 5′:3′ ratio was 1:1, the SHAPE reactivity of the UFS element in 5′-300 nt WT RNA exhibited a significant increase compared with the reactivity when the 3′ UTR was absent, and the SHAPE reactivity of the SLB loop and 5′ CS region was obviously reduced. When the 3′ UTR was in 5-fold excess of 5′-300 nt RNA, the SLB loop region of WT 5′ RNA showed essentially no SHAPE reactivity, indicating a complete transition from the SLB structure to a UAR duplex. In contrast, when the 5′-300 nt M2C RNA was incubated with an equal concentration of 3′ UTR RNA, there were only minor changes in the SHAPE reactivity of the UFS and SLB loop regions compared with those of 5′-300 nt M2C RNA alone. When the 5′:3′ ratio was 1:5, the SLB loop region of 5′-300 nt M2C RNA still showed moderate SHAPE reactivity. The above results demonstrated that the UFS duplex is melted when the 3′ UTR binds to 5′ end RNA, and stabilization of the UFS renders it difficult to unwind in response to the 3′ UTR and thus hinders genome cyclization.

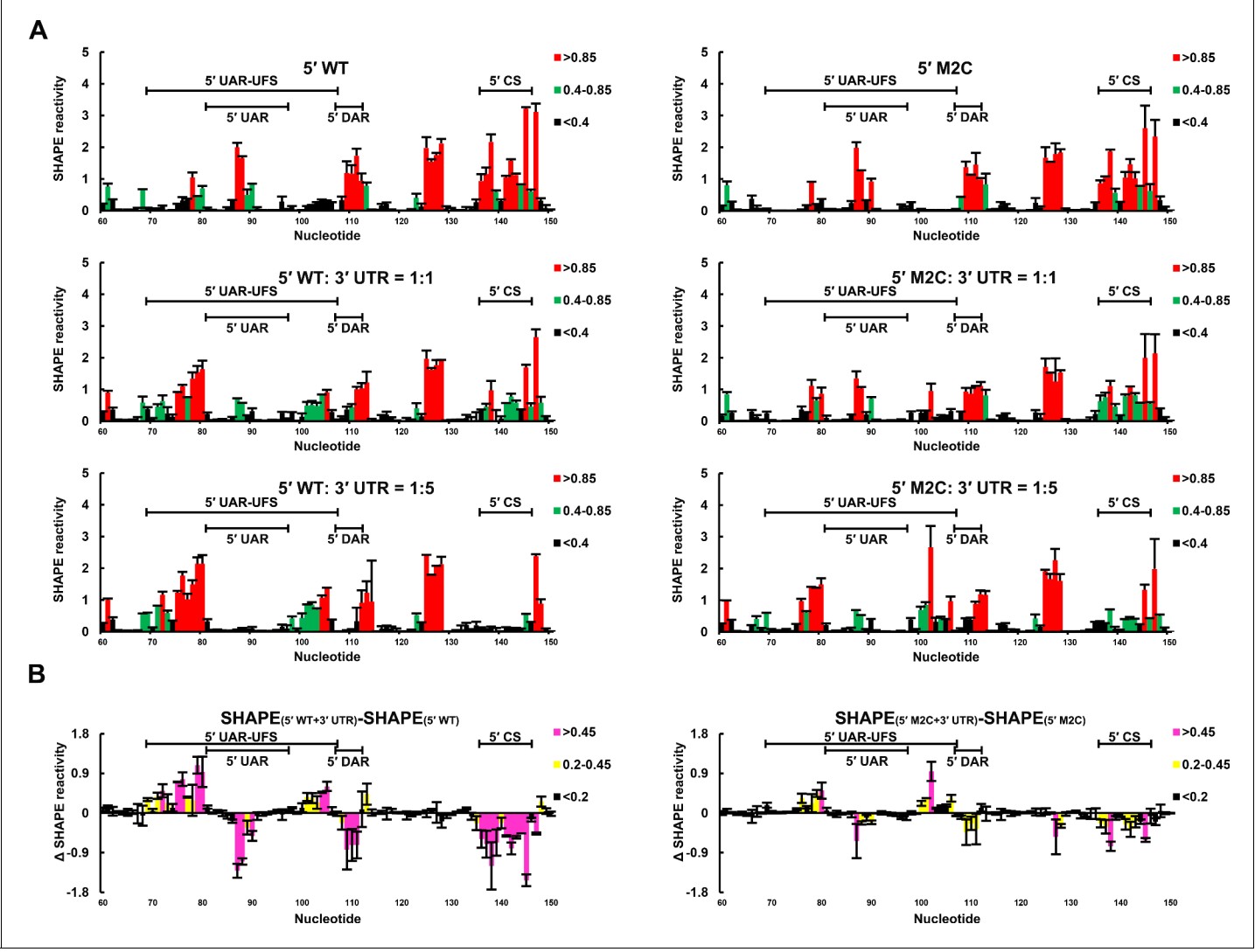

**Figure 9.** SHAPE analysis of the interactions between DENV4 genomic ends. (**A**) SHAPE analysis was performed for DENV4 5′-300 nt WT and M2C RNA in the presence of various amounts of 3′ UTR RNA. For convenience, only the results for nucleotides 60–150, which include critical elements for genome cyclization, are shown. The regions corresponding to various 5′ elements are indicated. Negative SHAPE reactivity is set to zero for clear demonstration. (**B**) The changes in the SHAPE reactivity of the 5′ 60–150 region were obtained by subtracting the SHAPE reactivity when 3′ UTR RNA was absent from the corresponding values when 3′ UTR RNA was present in a 1:1 ratio with 5′-300 nt RNA. The results were from two biological replicates, and the error bars represent the standard deviation.

The following source data is available for figure 9:

**Source data 1.** Source data for *Figure 9*.

Thus, although the UFS does not participate in 5′-3′ hybridization directly, the stability of its secondary structure must be maintained below a threshold to enable the necessary conformational changes required for genome cyclization. It can be speculated that the unwinding of the UFS during genome cyclization may be important for its function in viral RNA synthesis.

## Genome cyclization reduces the affinity of NS5 for vRNA by inducing the unwinding of the UFS

As the UFS duplex is unwound after genome cyclization, it should not functionalize on circularized genomes. To investigate this inference, artificial mini-genomes (*Figure 10A*), which have been widely

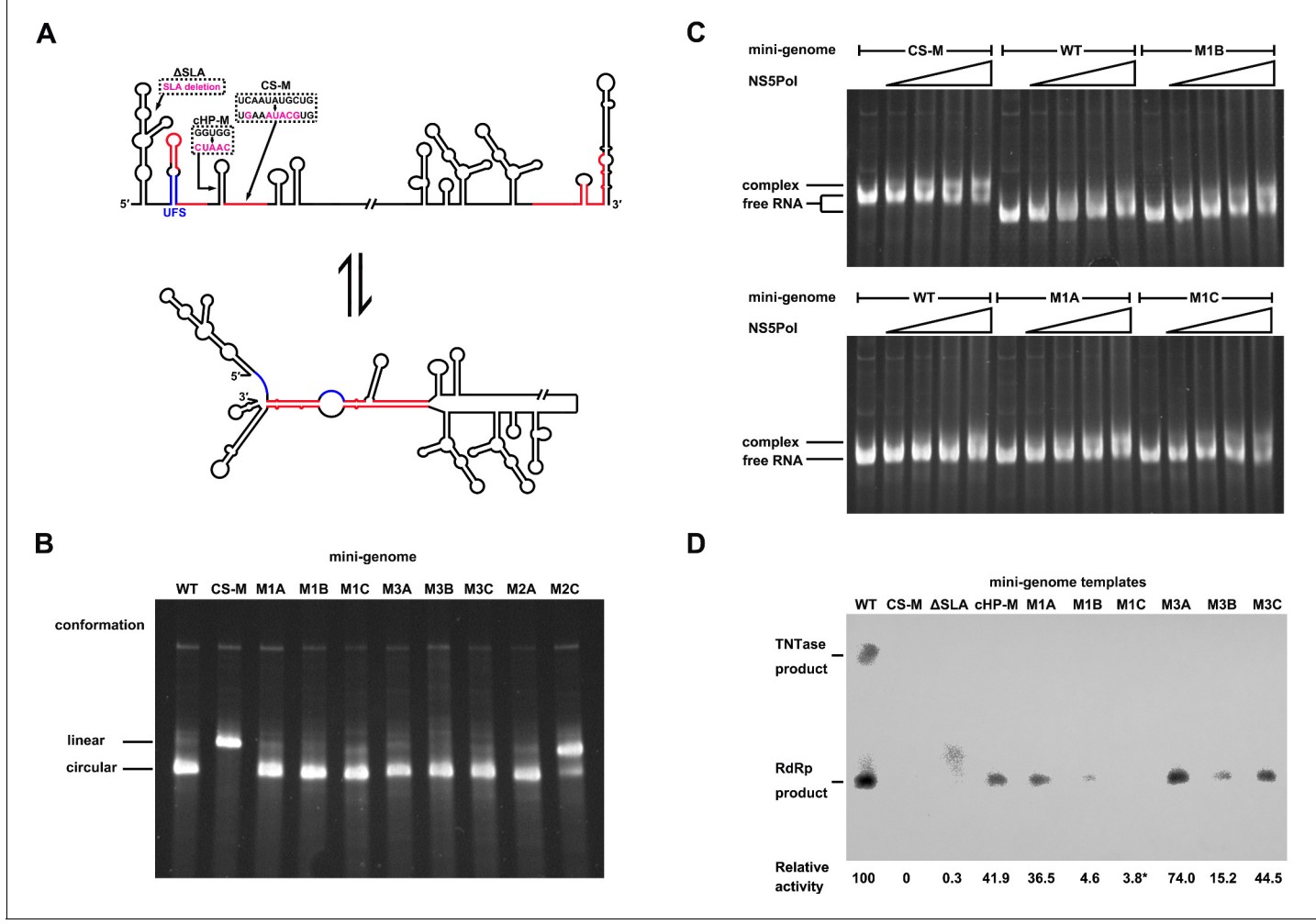

**Figure 10.** Genome cyclization disables the function of the UFS in NS5 recruitment. (**A**) Schematic diagram showing the conformational changes of the DENV4 genome. Two major conformations of vRNA, the linear and circular form, exist in equilibrium. The regions involved in terminal interactions are colored in red, and the UFS element is labeled in blue. The locations and sequences of the ΔSLA, cHP-M and CS-M mutants used in this section are indicated. (**B**) The conformational equilibrium of mini-genome RNA containing different mutations was analyzed using native PAGE. The bands of the linear and circular conformations were identified using the CS-M mutant as a control. The faint and much larger bands were likely to be formed by misfolded RNA molecules. (**C**) The binding of NS5Pol to different mini-genome RNA molecules was analyzed by EMSA. Mini-genomes containing the M1 series mutations in the UFS were assayed, and the WT and CS-M mini-genomes served as control reactions. No NS5Pol was present in the left first lane of each group. The NS5Pol concentrations in the reactions were approximately 2.0, 3.5, 5.0 and 7.5 μM (from left to right). (**D**) In vitro RdRp activity assay using various mini-genome RNA molecules as templates. The reactions were performed at 30°C for 40 min. Note that the signal of the ΔSLA lane did not correspond to the dsRNA product generated by de novo initiation. The blot was analyzed by ImageJ software and RdRp activity was expressed as percentage of WT. Note that the template activity of M1C-mini was calculated by analyzing a longer exposed version of the same blot shown.

used in studies of flavivirus vRNA synthesis, RdRp activity and RNA structures (*Niyomrattanakit et al., 2015*; *Potisopon et al., 2014*; *Sztuba-Solinska et al., 2013*; *You and Padmanabhan, 1999*), were employed. The DENV4 WT mini-genome and those containing the desired mutations were constructed (*Figure 10A*). First, using native PAGE, we found that the WT mini-genome, as well as the M1A, M1B, M1C, M3A, M3B, M3C and M2A mini-genomes, existed mostly in a circularized conformation (*Figure 10B*), whereas the CS-M mutant, in which the 5′ CS sequence was mutated, was confined to a linear conformation, as expected. The M2C mutant also mainly stayed in a linear conformation, agreeing with the results of the 5′-3′ RNA binding assay. The above results demonstrated that these corresponding mini-genomes are highly circularized; thus, it is unlikely that the UFS duplex structure exists in them. Next, the affinity of different mini-genomes for

NS5Pol was assessed. The WT, M1A, M1B and M1C mini-genomes all showed low levels of binding to NS5Pol (*Figure 10C*). In contrast, the CS-M mini-genome bound to NS5Pol with apparently higher affinity than the other mini-genomes tested (*Figure 10C*).

Next, an in vitro RdRp assay was performed using mini-genomes as templates. The CS-M mini-genome, together with a mini-genome lacking the SLA element (ΔSLA) or with a disrupted cHP structure (cHP-M), were assayed in parallel as controls, which showed non-detected or reduced RdRp activity for de novo initiation (*Figure 10D*), in agreement with their known defects in vRNA replication. The de novo products in reactions using M1 and M3 series mini-genomes as templates were also all reduced compared to those produced using the WT mini-genome as a template suggesting that the UFS is disabled in mini-genomes because of the unwinding of its duplex (*Figure 10D*). The differences in RdRp activity between WT mini-genome and the UFS mutants were likely caused by changes of primary sequence in these constructs, In supporting of this, the M1C and M3C mutants were not able to reach the same level of replication as WT in replicon assay (*Figure 3D*), and it has been reported that the base composition of U-rich sequence in the UFS was actually able to affect vRNA replication of DENV2 (*Friebe and Harris, 2010*).Taken together, the above results demonstrated that genome cyclization disables the function of the UFS in NS5Pol recruitment, suggesting that the UFS has to function dynamically during vRNA replication. Moreover, the attenuation of the affinity of NS5Pol for vRNA by genome cyclization is potentially important for the translocation of NS5Pol from the 5′ end to the 3′ end of vRNA.

## Mechanistic model for the function of the UFS in flavivirus replication

It has been suggested that both the linear and circular conformations of the flavivirus genome are required for vRNA replication and exist in equilibrium (*Villordo et al., 2010*). However, circular states should be dominant among its conformations. Although the highly circularized mini-genome is much shorter than the actual viral genome, it has been theoretically proposed that the generic properties of a large RNA molecule lead its two ends to stay close to each other (*Yoffe et al., 2011*), which was supported by results from single-molecule FRET assays (*Leija-Martinez et al., 2014*). Thus, the 5′ and 3′ ends of the flavivirus genome have great opportunities to contact each other, and due to the high affinity between them, most vRNA molecules would stay circularized in the equilibrium state. Moreover, protein factors have been shown to facilitate the cyclization of mRNA *and* vRNA in vivo. In fact, the PABP protein, which bridges mRNA ends through poly(A)-PABP-eIF4E-5′ cap interactions, has been shown to bind to the DENV 3′ UTR (*Polacek et al., 2009*). However, if the linear conformation of vRNA, the only conformation under which the UFS can exist, is present in low abundance, the question remains of how the latter can function.

Inspired by the co-transcriptional folding of RNA molecules (*Frieda and Block, 2012*; *Gong et al., 2015*; *Meyer and Miklos, 2004*; *Perdrizet et al., 2012*), we speculated that the UFS duplex is folded during the ongoing synthesis of nascent positive-strand RNA because the 5′ end of the parental (+) strand is released from the dsRNA replication form (RF) by a potential strand-displacement mechanism during the process, given that the flavivirus genus exhibits semi-conservative replication (*Chu and Westaway, 1985*; *Uchil and Satchidanandam, 2003*), whereas the 3′ end of the parental (+) strand RNA remains in double-stranded form with the (−) strand.

Based on the above considerations, we proposed that the UFS functions as an RNA switch, and its 'ON' and 'OFF' states are controlled by genome cyclization (*Figure 11*) to fulfill different needs in vRNA replication. A mechanistic model to explain the function of the UFS is described below. (i) vRNA released in the cytoplasm is translated to generate a sufficient level of viral proteins, which leads to the eventual binding of NS5 to the SLA and the initiation of (−) strand synthesis, even without the presence of the UFS duplex. (ii) Completion of (−) strand synthesis generates the dsRNA RF, which serves as the template for the synthesis of progeny vRNA in the presence of viral NS5/NS3 and other viral/host factors (not shown in *Figure 11*). (iii) During the elongation of the nascent (+) strand, the 5′ terminus of the parental (+) strand RNA is displaced first, and the UFS duplex is folded on the protruding 5′ end; this 'ON' state of the UFS facilitates the recruitment of NS5 to the free 5′ end. (iv) After the completion of the nascent chain synthesis, the NS5-bound vRNA undergoes genome cyclization, and the UFS element is set to 'OFF' by the hybridization between genome termini, which decreases the binding strength of NS5 to the 5′ terminus and promotes its translocation to the 3′ terminus to initiate next-round synthesis of negative-strand vRNA. Such dynamic modulation of NS5 binding should be of great significance to maintain an appropriate level of (−) vRNA

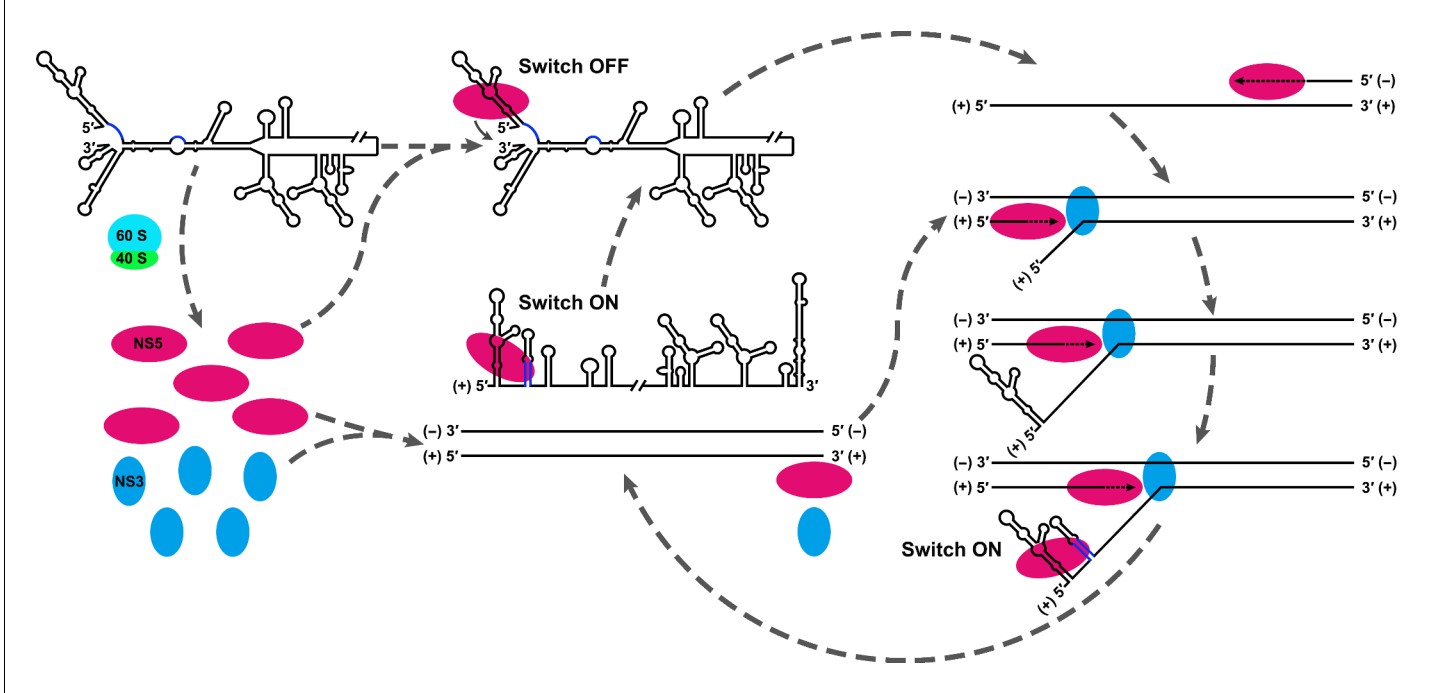

**Figure 11.** Model explaining the functional mechanism of the UFS switch. A proposed mechanistic model of flavivirus vRNA replication. (i) After the viral genome is released into the cytoplasm, first, translation occurs on the circularized genome to generate a sufficient level of viral proteins for downstream vRNA synthesis. (ii) Accumulation of viral replication proteins (only NS5 and NS3 are shown) to high concentrations results in the formation of an RNA-viral replicase complex and initiates (−) RNA synthesis to generate the double-stranded replication form (RF). (iii) Then, the synthesis of nascent (+) RNA using RF as the template is initiated by NS5/viral replicase. During the process, 5′ local structures including UFS are formed in the displaced, parental (+) strand RNA. A free NS5/viral replicase is recruited to bind the 5′ end of the displaced (+) RNA. (iv) As soon as the nascent (+) strand RNA synthesis completes, the released (+) ssRNA genome is circularized by terminal interactions, and the next round of (−) strand synthesis is ready to start.

synthesis to ensure the global balance of vRNA replication because the dsRNA RF is a highly active template for (+) vRNA synthesis.

Furthermore, this RNA-switching mechanism is likely to be universal among flaviviruses with vertebrate hosts given that when the terminal secondary structures of both genome conformations of different flaviviruses were examined, we found that the 5′ local structures immediately following the SLA elements were always melted in response to genome cyclization (*Figure 12* and *Figure 12—figure supplements 1–4*), suggesting that these structures may possibly possess the same regulatory function as the UFS.

## Discussion

Herein, via the combination of bioinformatics, biochemical and reverse genetics approaches, the UFS element was shown to be crucial for efficient flavivirus vRNA replication through dynamic modulation of viral RdRp recruitment. Our results demonstrated that the UFS links two events of vRNA replication, NS5 recruitment and genome cyclization, together, and the structural stability of the UFS is elaborately balanced to meet the requirements of both. Especially, the conformational change of the UFS caused by genome cyclization is coupled with its function in modulating dynamic RdRp recruitment. As the function of UFS in viral replication and RdRp recruitment was investigated using different flaviviruses (DENV4, JEV and ZIKV), it is plausible to consider that the UFS is a general regulatory strategy of viral replication among the flavivirus genus. Although a previous report (*Filomatori et al., 2011*) has suggested that the UFS may not be functional in DENV2, this was more likely to be an exception, as our SHAPE data also indicated that the UFS in DENV2 is highly unstable in vitro, in contrast to the UFS elements from other viruses (*Figure 2*).

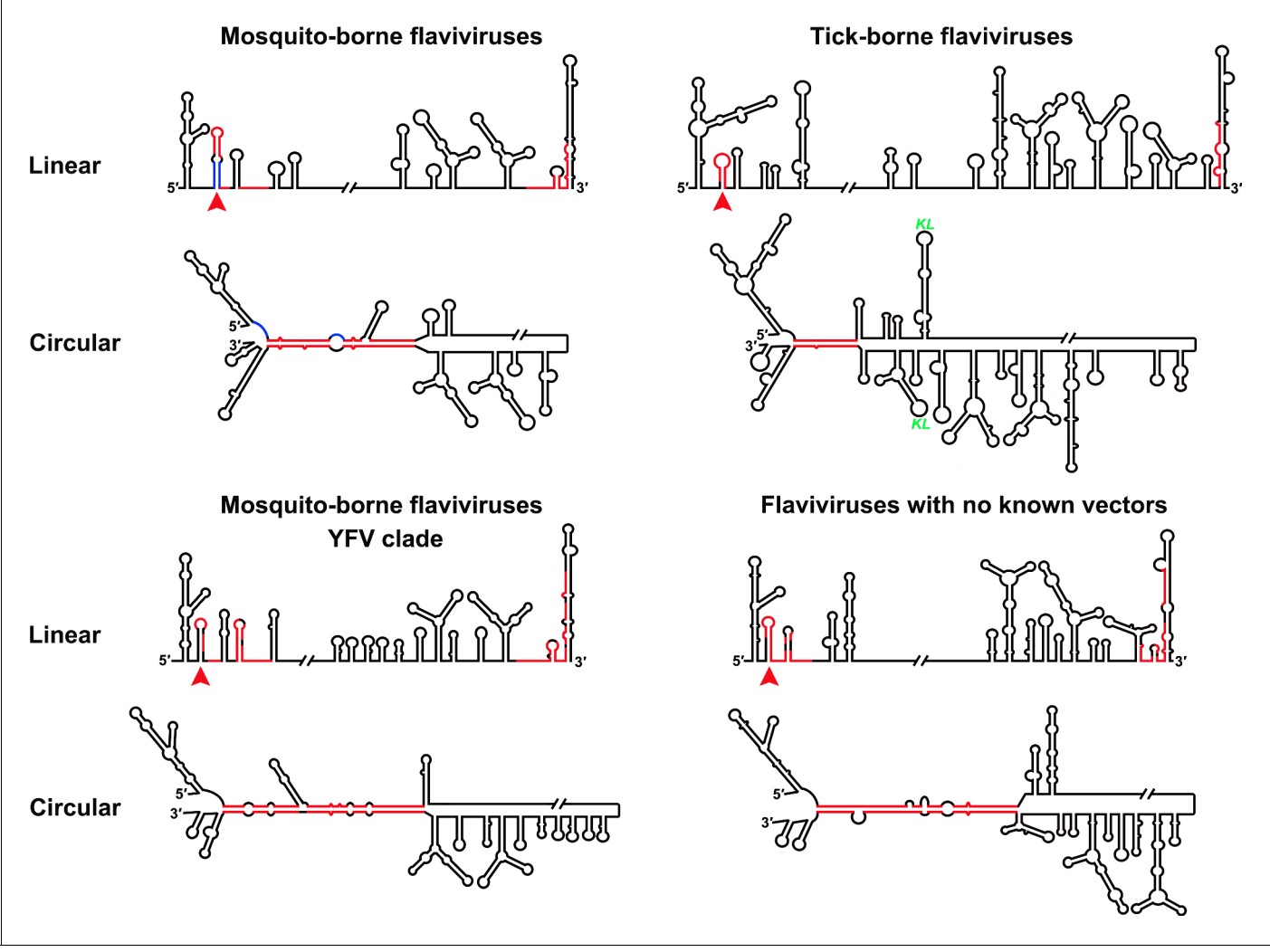

**Figure 12.** Genome conformation models of major groups of flaviviruses. Sequences involved in terminal interactions are shown in red. The 5′ RNA structures that are immediately downstream of SLA elements and consistently involved in genome cyclization are indicated by red arrows. The typical UFS element in MBFVs is shown in blue. The demonstrated structures are based on DENV4 for the MBFVs; tick-borne encephalitis virus (TBEV) for TBFVs; Wesselsbron virus (WESSV) for the YFV clade of the MBFVs and RBV for the NKVs. The kissing loop interaction identified recently in the tick-borne group (*Tsetsarkin et al., 2016*) is labeled by green letters (KL).

The following figure supplements are available for figure 12:

**Figure supplement 1.** Terminal genomic RNA structures of the MBFV group.

**Figure supplement 2.** Terminal genomic RNA structures of the YFV clade in the MBFV group.

**Figure supplement 3.** Terminal genomic RNA structures of the TBFV group.

**Figure supplement 4.** Terminal genomic RNA structures of the NKV group.

By applying a systemic RNA structure prediction and comparison approach covering different members of the flavivirus genus, the structure of the downstream SLA region in the 5′ end of the flavivirus genome was unambiguously illuminated. The UA-base-pair-rich UFS and the UFS-like element have been identified in MBFVs, two clades of NKVs, and ISFVs, demonstrating that the UFS element is evolutionarily conserved. Interestingly, while the UFS duplex unwinds during genome cyclization in MBFVs and NKVs, neither the conserved mode of genome cyclization nor the unwinding of UFS-like

elements during this process was found in ISFVs. The ISFVs have been indicated to be the most divergent outgroup among the genus flavivirus (*Blitvich and Firth, 2015*; *Cook and Holmes, 2006*; *Moureau et al., 2015*), which led us to assume that modern flaviviruses are evolved from an insect virus ancestor. We speculated that primordial UFS-like structures are potential viral replication enhancers with RdRp-recruiting functions in ancient ISFVs. When the viruses were introduced into vertebrates, the intense selection pressure caused by shifting between two hosts promoted the emergence of a highly efficient genome cyclization strategy, and the UFS-like elements evolved into the RNA switches in modern flaviviruses. Moreover, although the UFS structure itself was lost when the ancestral ISFV adapted into ticks and some special vertebrates, its functional mechanism was inherited by the appearance of elements such as 5′-SL2/CS-A in TBFVs. Thus, by looking into the evolutionary history of flaviviruses and exploring the modes of genome cyclization among them, we propose that the UFS may play a critical role during the evolution of the genus flavivirus.

UFS elements usually localize close to, or overlap with, the translation start region, suggesting a possible role for the UFS in viral translation regulation. Moreover, the replication of most positive-strand RNA viruses occurs in virus-induced membrane structures (*Gillespie et al., 2010*; *Welsch et al., 2009*). It is possible that during (+) vRNA synthesis of flaviviruses, the parental (+) RNA is anchored to the vesicle membranes and prevented from being transported out of the vesicle through interactions between the UFS-containing 5′ end and the membrane-bound viral replicase. Interestingly, DENV has been reported to conceal its dsRNA in membrane structures to escape from host immunity surveillance (*Uchida et al., 2014*). Thus, other potential functions of the UFS need further investigation.

Functional long-range interactions are likely to be a general mechanism in positive-strand RNA viruses (*Nicholson and White, 2014*). Genome cyclization is responsible for the delivering of viral replication factors that bind 5′-terminally or internally to the very 3′ end of the viral genome. Such tactics have been observed in positive-strand RNA viruses, which infect animals, plants and bacteria (*Nicholson and White, 2014*). However, to our knowledge, this is the first report describing that an RNA switch triggered by genome cyclization integrates the recruitment and translocation of the viral RdRp into the dynamic process of nascent RNA synthesis. This deliberate control mechanism used by flaviviruses could also be used by other positive-strand RNA viruses in similar forms, and further pursuit of these mechanisms will provide deeper insight into the replication and evolution of positive-strand RNA viruses.

## Materials and methods

### DNA constructs

The p4 infectious clone (*Durbin et al., 2001*) of the DENV4 814669 strain and its host *E. coli* strain BD1528 were kindly provided by Professor Stephen S. Whitehead. All replicon constructs were also manipulated using BD1528. The p4-Dualstop-SP-IRES-Rluc-Rep replicon was generated by introducing a nonsense mutation into the 26th codon of the capsid ORF (CAA to TAA) in the background of p4-cHPstop-SP-IRES-Rluc-Rep (*Liu et al., 2013*). p4-Dualstop-SP-IRES-Rluc-Rep-GVD, in which the NS5 catalytic triad was mutated, was generated based on p4-cHPstop-SP-IRES-Rluc-Rep-GVD using similar strategies.

For the convenience of introducing mutations into the UFS region, the Asc I-SnaB I fragment from p4-Dualstop-SP-IRES-Rluc-Rep, which contained the SP6 promoter region and the DENV4 5′ UTR-capsid ORF sequence, was engineered such that the sequence corresponding to the 70–107 nt region of the DENV4 genome was substituted by the following: 5′- <u>CTCCTC</u>ACACATACGTAA TGGG<u>GAGGAG</u>-3′. This engineered fragment contained two inversely positioned BseR I sites (underlined above) and was cloned into pGEM-T easy (Promega, Madison, Wisconsin) to generate the cloning cassette vector, pBseRI-Dualstop-04. DNA oligos containing the desired mutations of the UFS were annealed and ligated with BseR I-digested pBseRI-Dualstop-04 to acquire Asc I-SnaB I fragments with the desired mutations, which were then subcloned back into p4-Dualstop-SP-IRES-Rluc-Rep.

To generate UFS mutants in the p4 infectious clone and p4-GVD, the Asc I-BstE II fragments with desired mutations were generated by overlapping PCR using the corresponding replicon mutants

and p4 infectious clone DNA as templates for amplification, cloned into pGEM-T easy, and then subcloned into p4 and p4-GVD, respectively.

Site-directed mutagenesis method was employed for the generation of ZIKV fragments containing the desired UFS mutations, the generated Not I-Avr II restriction fragments were then subcloned into the infectious clone of ZIKV strain FSS13025 (*Shan et al., 2016*), which was kindly provided by Professor Pei-Yong Shi.

The DENV4 mini-genomes were constructed by joining various 5′-300 nt fragments with the 3′ UTR sequence using overlapping PCR techniques. The mutations targeting cHP and 5′CS were also introduced by overlapping PCR.

The DENV2 and DENV3 5′ end sequences were acquired by RT-PCR using the DENV2 strain NGC (KM204118) and the DENV3 strain 80–2 (AF317645) RNA as templates respectively, and cloned into pMD-19T-simple (Takara, Dalian, China). The 5′ end sequence of DENV1 strain WestPac (U88535) was chemically synthesized and cloned into pUC57 by Thermo Fisher Scientific. The SP6 promoter sequence was placed upstream of the DENV1 5′ end sequence. These constructs, together with the JEV infectious clone pAJE70 (*Li et al., 2013*), which contains the 5′ UTR-C-coding region of strain SA-14-14-2 (AF315119), were utilized to amplify the DNA templates for in vitro transcription. The infectious clone of ZIKV strain FSS13025 was used to amplify the DNA template for in vitro transcription of ZIKV 5′ end RNA. JEV UFS mutants were generated by site-directed mutagenesis, the resultant plasmids were utilized to amplify the DNA templates for in vitro transcription of the corresponding JEV 5′ RNA mutants.

## RNA preparation

Replicon and full-length viral RNA were in vitro transcribed as described previously (*Liu et al., 2013*). To generate RNA species for in vitro SHAPE, EMSA and RdRp activity assays, high-fidelity PCR-amplified DNA fragments, which contained the SP6 or T7 promoter, were used as templates for in vitro transcription reactions. RNA was purified with an RNeasy mini kit (Qiagen, Hilden, Germany), and the concentrations were determined spectrophotometrically. Agarose/TAE gel electrophoresis was routinely conducted to examine the integrity of the purified RNA. The RNA preparations were stored at −80°C before use. Note that all DENV4 5′ RNA and mini-genome RNA molecules contained two artificial nonsense mutations at the corresponding sites in the capsid coding region that have been shown not to affect the folding of the corresponding RNA or vRNA replication (*Clyde et al., 2008*; *Liu et al., 2013*).

## Replicon assay

For the evaluation of the replication characteristics of the p4-cHPstop-SP-IRES-Rluc-Rep and p4-Dualstop-SP-IRES-Rluc-Rep replicons, a replicon assay was performed as described previously (*Liu et al., 2013*). For the replicon assays of the UFS mutants, ten thousand BHK-21 cells were seeded into each well of Nunc MicroWell 96-Well Optical-Bottom Plates (Thermo Fisher Scientific, Waltham, Massachusetts) one day prior to transfection. Unless otherwise specified, 100 ng of replicon RNA was transfected per well using Lipofectamine 2000 reagent (Thermo Fisher Scientific, Waltham, Massachusetts), and the plates were cultured at 37°C with 5% $CO_2$. At the indicated time points, the cells in the microplates were lysed with 1× *Renilla* lysis buffer (Promega, Madison, Wisconsin) and stored at -20°C until use. The Renilla luciferase activity of the collected samples was measured as described previously (*Liu et al., 2013*), except that the microplates were directly subjected to the GloMax-96 luminometer.

## Electroporation of full-length vRNA, qRT-PCR and virus titration

Four million BHK-21 cells suspended in 400 µl of Opti-MEM I medium were electroporated with 5 µg of vRNA. The details of the electroporation are described elsewhere (*Liu et al., 2013*). The electroporated cells were resuspended in 14 ml of Dulbecco's modified minimal essential medium (DMEM, Thermo Fisher Scientific, Waltham, Massachusetts) with 5% fetal bovine serum (FBS, PAA, Morningside, QLD, Australia) and dispensed into 24-well plates with or without sterile cover slips. The cells were cultured at 37°C in 5% $CO_2$. At 4 hr post-electroporation, a media change was performed for the plates intended for vRNA quantification analysis. At the indicated time points, the total RNA was isolated from the transfected cells, and qRT-PCR was performed to quantify vRNA

copies as described previously (*Liu et al., 2013*). The culture supernatants from the transfected cells were collected at different time points post-transfection, and virus titers were determined by plaque-forming assays.

## Indirect immunofluorescence assay

BHK-21 cells electroporated with corresponding vRNA were seeded onto cover slips in 24-well plate-format and cultured at 37°C in 5% $CO_2$. At the indicated time points, the cover slips were washed once with PBS (phosphate buffered saline, pH 7.4) and fixed with acetone/methanol (v/v: 3/7) at 4°C. The cover slips were incubated with the anti-dengue envelope protein monoclonal antibody 2A10G6 at 37°C for 1 hr for the cells transfected with DENV4 RNA, and anti-ZIKV envelope protein mAb clone 0302156 (1:1000 diluted, BioFront Technologies, Tallahassee, Florida, USA) was used for the cells transfected with ZIKV RNA. After the incubation with primary antibodies, the cover slips were washed with PBS for three times. AlexaFluor 488-labeled goat anti-mouse IgG (1:200 diluted, zsbio, Beijing, China) was then added, and after a 1-h incubation, the cover slips were washed as described above. For cell nuclei staining, 4', 6-diamidino-2-phenylindole (DAPI, 0.5 ng/µl) was added onto the cover slips and incubated for 5 min. An Olympus BX51 microscope under the control of DP72 software was utilized for image capture.

## SHAPE analysis

RNA molecules (approximately 9.66 pmol) corresponding to the WT or mutated 5' end sequence of different flaviviruses were diluted into 12-µl reactions containing 0.5× TE buffer, heated at 95°C for 2 min and snap-cooled on ice immediately. Then 6 µl of 3.3× RNA folding buffer (333 mM HEPES, pH 8.0, 333 mM NaCl and 20 mM $MgCl_2$, *Wilkinson et al., 2006*) was added and the RNA was refolded at 37°C for 20 min. The 18-µl refolded RNA was splitted equally into two tubes and then modified by N-methylisatoic anhydride (NMIA, Sigma-Aldrich, St. Louis, Missouri). For the analysis of 5' end RNA-3' UTR complexes, the two RNA segments were co-folded at the indicated molar ratios and then subjected to NMIA modification. The SHAPE (+) reactions contained 1 µl of 130 mM NMIA freshly dissolved in dimethyl sulfoxide (DMSO, Sigma-Aldrich, St. Louis, Missouri), and 1 µl of DMSO was added into the parallel SHAPE (−) reactions. The SHAPE (+) and (−) reactions were incubated at 37°C for 45 min, then the RNA in the reactions was purified using RNA Clean and Concentrator-5 (Zymo Research, Irvine, California, USA). Primer extension reactions utilizing fluorophore-labeled primers were performed with Superscript II reverse transcriptase (Thermo Fisher Scientific, Waltham, Massachusetts) in a 20 µl volume according to the manufacturer's instructions. ddTTP dideoxy-sequencing reactions were performed in parallel. We routinely used VIC-labeled primers for primer extension of the SHAPE reactions and NED-labeled primers for sequencing reactions. Otherwise, FAM- and HEX-labeled primers were utilized for primer extension of the SHAPE and sequencing reactions, respectively. After primer extension, one microliter of 4 M NaOH was added to each reaction, and treatment at 95°C for 3 min was performed to hydrolyze the RNA templates. Two microliters of 2 M HCl were added to adjust the pH to neutral. Then, the sequencing reactions were split and added into the SHAPE (+) and (−) reactions. The final mixtures were purified by ethanol/EDTA precipitation, dissolved in Hi-Di formamide and separated via denaturing PAGE capillary electrophoresis by the DNA sequencing department of Thermo Fisher Scientific. The SHAPE data were analyzed using QuShape software (*Karabiber et al., 2013*). All the processing steps were performed using the default software settings. Negative SHAPE reactivity was set to zero except for the calculations of the Δ SHAPE reactivity of WT and M2C 5'-300 nt RNA in response to 3' UTR RNA.

## Expression and purification of recombinant DENV4/JEV NS5Pol

The coding sequence of the DENV4 NS5 RdRp domain (residues 270–900, NS5Pol) was cloned into a pET-28a (+) expression vector using Nhe I/Xho I restriction sites. The recombinant NS5Pol carrying an N-terminal His-tag was expressed in *E. coli* Rosetta (λDE3) in the presence of 0.125 mM isopropyl-β-D-thiogalactopyranoside (IPTG) at 16°C for 20 hr. The cell pellets were harvested and washed once with Buffer A (25 mM Tris.Cl, pH 8.0, 500 mM NaCl, 25 mM imidazole and 5% glycerol) and resuspended in Buffer A containing cOmplete EDTA-free protease inhibitor cocktail tablets (Roche, Penzberg, Germany). After ultra-sonication, the suspension was centrifuged at 30,000 g at 4°C for 20 min. The soluble recombinant NS5Pol protein in the supernatants was then purified using a

HisTrap HP column (GE Healthcare, Little Chalfont, UK). The fraction containing NS5Pol was eluted with Buffer A containing 300 mM imidazole. The eluted protein was exchanged into 50 mM Tris.Cl (pH 8.0), 300 mM NaCl and 20% glycerol using a HiTrap desalting column (GE Healthcare, Little Chalfont, UK) and concentrated to mg/ml grade by ultrafiltration using Amicon Ultra-15 Centrifugal Filter Units (30 kDa-cutoff, Merck Millipore, Billerica, Massachusetts, USA). Dithiothreitol (DTT) was added into the purified NS5Pol protein to a final concentration of 1 mM. The final product was dispensed into single-use aliquots and stored at -80°C before use. NS5Pol of JEV SA-14-14-2 strain was cloned, expressed and purified using the same strategy.

## EMSA assay

An RNA-RNA binding EMSA was performed as described previously (*Liu et al., 2013*) with minor modifications. For the NS5Pol-RNA EMSA assays, the different DENV4 5'-300 nt RNA species were first diluted in 0.5× TE buffer and heated at 95°C for 2 min. The RNA samples were then placed on ice immediately. 5× RNA folding buffer T (250 mM Tris.Cl, pH 8.0, 500 mM NaCl and 25 mM MgCl$_2$) was added to the samples, and the RNA was refolded at 37°C for 20 min. The concentration of the renatured RNA was set to 500 nM. The binding reactions contained 50 nM 5'-300 nt RNA, 4 µl of 5× EMSA buffer (200 mM Tris.Cl, pH 8.0, 300 mM NaCl, 25 mM MgCl$_2$), 0.05 mg/ml heparin sodium salt, 7.5% glycerol and different amounts of NS5Pol (0, 3, 5.25, 7.5 and 11.25 µg) in a 20 µl volume. Due to the solvent components in RNA and protein preparations, the final reactions contained 64.15 mM Tris.Cl (pH 8.0), 182.5 mM NaCl, 5.5 mM MgCl$_2$, 40 nM EDTA and 375 nM DTT. The reactions were incubated at 30°C for 30 min, and then, 10× gel loading solution (Thermo Fisher Scientific, Waltham, Massachusetts) was added, and the mixtures were separated by electrophoresis on 6% native PAGE gels running in 0.5× TBE. The gel apparatus was placed in an ice-water bath to prevent the dissociation of RNA-protein complexes. After running at 60 V for 4 hr, the gels were stained with SYBR Gold nucleic acid gel stain (Thermo Fisher Scientific, Waltham, Massachusetts) for 30 min, and gel pictures were captured using a BioSpectrum Imaging System (UVP, LLC, Upland, California) with a UV transilluminator. EMSAs using 5'-160 nt RNA or mini-genome RNA were performed similarly, with a few exceptions: for the 5'-160 nt RNA-based EMSA, the NS5Pol concentrations were higher (0, 5.4, 10.8, 16.2 and 21.6 µg, respectively); for the mini-genome-based EMSA, the heparin concentration was set to 0.16 mg/ml, and the electrophoresis was performed with 4.5% PAGE gels at 75 V for 6 hr in an ice-water bath. EMSA assay using JEV NS5Pol and 5'-320 nt RNA was performed with similar conditions for DENV4 5'-300 nt RNA EMSA assay.

## RdRp assay

In vitro RdRp initiation assays were performed using: (I) 5'-160 nt RNA of DENV4, (II) DENV4 mini-genome RNA as templates. The RNA templates were diluted with RNase-free water to a final volume of 8 µl, then heated at 95°C for 2 min and immediately cooled on ice. Two microliters of RNA fold buffer T2 (250 mM Tris.Cl, pH 7.4, 100 mM NaCl, 25 mM MgCl$_2$) was added, and the RNA templates were folded at 37°C for 20 min. Unless otherwise specified, the 30-µl RdRp reactions contained 450 ng of template RNA (3 µl of 150 ng/µl folded RNA), 55.6 mM Tris.Cl (pH 7.4), 12 mM NaCl, 5.5 mM MgCl$_2$, 2 mM MnCl$_2$, 10 mM DTT, 40 U of recombinant RNase inhibitor (40 U/µl), approximately 5.4 µg NS5Pol (in 1 µl) and 0.5 mM ATP, GTP, and UTP; 0.1 mM CTP; and 0.25 mM biotin-11-CTP (Roche, Penzberg, Germany). According to previous reports (*Ackermann and Padmanabhan, 2001*; *Filomatori et al., 2006*), the RdRp reactions were incubated at 30°C for the desired length of time. The reaction products were purified using an RNAclean Kit (Tiangen, Beijing, China), and 2× RNA loading dye (New England Biolabs, Ipswich, Massachusetts) was added into the purified RNA. The samples were preheated at 70°C for 5 min, snap-cooled on ice, and analyzed using 6% PAGE/8 M urea gels. The gel-separated RNA was electro-transferred onto a Hybond N+ Nylon membrane (GE Healthcare, Little Chalfont, UK). After air-drying the membrane, the transferred RNA was UV-crosslinked by a UV transilluminator for 10 min. The membrane was blocked using 10% nonfat dry milk, and then streptavidin-conjugated HRP (Thermo Fisher Scientific, Waltham, Massachusetts, USA) was incubated with the membrane at room temperature for 2 hr. The membrane was then extensively washed using PBS with 0.05% Tween 20, and chemiluminescence signals were detected using a Pro-light chemiluminescent kit (Tiangen, Beijing, China) and a Smartchemi II imaging system (Sage Creation Science Co, Beijing, China).

## Statistical analysis

Statistical analysis was performed using the statistical functions of GraphPad Prism 6 (GraphPad Software, La Jolla, California, USA). Briefly, Paired t-test was performed for *Figure 3B*. One-way ANOVA and Dunnett's multiple comparison test were performed for *Figure 2—figure supplement 1*, *Figure 3D*, *Figure 4*, *Figure 7* and *Figure 7—figure supplement 1*. Two-way ANOVA and Tukey's multiple comparison test were performed for *Figure 5B,C,D,F,G,H and I*. The detailed results of statistical analysis were provided within the corresponding source data files.

## Acknowledgement

We appreciate Prof. Stephen S Whitehead (National Institute of Allergy and Infectious Diseases, NIH, USA) and Prof. Pei-Yong Shi (University of Texas Medical Branch, USA) for the p4 infectious clone and the infectious clone of ZIKV strain FSS13025, respectively. The authors thank Dr. Guang-Chuan Wang (Zhejiang University) for insightful comments and discussions. This work was supported by the National Key Research and Development Project of China (2016YFD0500304), the National Natural Science Foundation of China (No. 31270196, 31000083 and 30972613) and National 973 project of China (2012CB518904). CF Qin was supported by the Excellent Young Scientist Program from the National Natural Science Foundation of China (81522025) and the Newton Advanced Fellowship from the UK Academy of Medical Sciences and NSFC (No. 81661130162).

## Additional information

### Funding

| Funder | Grant reference number | Author |
|---|---|---|
| National Natural Science Foundation of China | 31270196 | Cheng-Feng Qin |
| National Natural Science Foundation of China | 31000083 | Xiao-Feng Li |
| National Natural Science Foundation of China | 30972613 | Cheng-Feng Qin |
| National Natural Science Foundation of China | National Basic Research Program of China, 2012CB518904 | Cheng-Feng Qin |
| National Natural Science Foundation of China | Excellent Young Scientist Program, 81522025 | Cheng-Feng Qin |
| Academy of Medical Sciences | Newton Advanced Fellowship, 81661130162 | Cheng-Feng Qin |
| National Key Research and Development Project of China | 2016YFD0500304 | Cheng-Feng Qin |

The funders had no role in study design, data collection and interpretation, or the decision to submit the work for publication.

### Author contributions

Z-YL, Conception and design, Acquisition of data, Analysis and interpretation of data, Drafting or revising the article; X-FL, TJ, Analysis and interpretation of data; Y-QD, QY, HZ, J-YY, Acquisition of data; C-FQ, Conception and design, Analysis and interpretation of data, Drafting or revising the article

### Author ORCIDs

Cheng-Feng Qin, http://orcid.org/0000-0002-0632-2807

## Additional files

### Supplementary files

• Supplementary file 1. *Mfold* prediction for the terminal RNA structures of different flaviviruses.

• Supplementary file 2. *Mfold* prediction for the 5′ end RNA structures of yokose virus clade of NKV group.

• Supplementary file 3. *Mfold* prediction for analysis of the influence of DENV4 UFS mutations on the overall genome terminal RNA structures.

• Supplementary file 4. Structure models of DENV4 5′ WT, M1A, M1C, M3A and M3C RNA generated by *RNAstructure* software using SHAPE constraints.

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
