## [Decision Letter]

Thank you for submitting your article "Viral RNA switch mediates the dynamic control of flavivirus replicase recruitment by genome cyclization" for consideration by *eLife*. Your article has been reviewed by three peer reviewers, and the evaluation has been overseen by a Reviewing Editor and Wenhui Li as the Senior Editor. The following individuals involved in review of your submission have agreed to reveal their identity: Jeffrey S Kieft (Reviewer #1) and Alexander Khromykh (Reviewer #3).

The reviewers have discussed the reviews with one another and the Reviewing Editor has drafted this decision to help you prepare a revised submission.

Summary:

Liu et al. propose that a previously undetected structural element in the 5' UTR of diverse flaviviruses is important for replication, but that upon genome cyclization this structure is disrupted by 5'-3' interactions. They propose that the RNA acts as a structural switch to coordinate cyclization and RdRP binding. Overall, the reviewers found that the model presented is reasonable but is not adequately supported by the experimental results. If the authors can provide additional data, analysis, and discussion to support this model, the manuscript may become suitable for publication in *eLife*.

Essential revisions:

1) The authors used many mutants, taking into account the impact on the linear form of the genome. However, the authors must also consider and discuss the impact of these mutations on RNA cyclization and on the internal loop structure (including UFS sequences) in the circular form. The predicted cyclization of DENV4 RNA is similar to that reported for JEV (Khromykh et al. 2003 JVI Figure 1), and different from other dengue serotypes. The model of genome cyclization of DENV4 RNA must be revised to include account possible hybridization of the U rich region of UFS with 3' end sequences (Figure 3, Figure 5 and Figure 7). For instance, in the M1 series, M1A may increase complementarity while M1B could reduce complementarity, similar analysis can be done for the M3 series. This possibility should be taken into account for the interpretation of results with all the mutants and the sequence/structure changes of the internal loop in the mutants should be analyzed.

2) The data shown in Figure 4 indicate that mutations within UFS alter SHAPE reactivity of SLA (which is the promoter for polymerase binding and activity), possibly explaining the reduced binding of the polymerase in the UFS mutants. Evaluation of the impact of UFS mutations on the overall molecule is crucial because the results regarding NS5 binding and RNA synthesis may be due to undesired changes within the promoter SLA. The authors must use the SHAPE data to generate predictions for these changes, for instance using RNAstructure software.

3) The authors find that NS5 affinity for the RNA is reduced when the UFS structure is disrupted by mutations. Considering that this observation is not caused by changes in SLA, the results do not agree with previous observations reported for DENV2. Specifically, mutations, deletions or complete replacement of UFS by unrelated sequences did not affect NS5 binding nor RNA synthesis in vitro (Filomatori et al. 2011 JBC Figure 2 and Figure 5, and Lodeiro et al. JVirol. 83,2, 993, 2009). Importantly, a substitution of the bottom half of UFS by As in the DENV2 infectious clone (mutant 6U/6A, Lodeiro et al. JV 2009) results in a replication competent virus similar to the WT. In this case UFS is completely disrupted and the virus shows efficient replication. These discrepancies suggest that the observations reported here are not applicable to other flaviviruses. The authors must better present their work in light of these potential differences; if the authors wish to conclude that their findings can be extended to other flaviviruses, additional data must to be provided to support this.

4) The authors propose that UFS formation is necessary for NS5 recruitment (linear form of the RNA) and when cyclization takes place, NS5 is released, promoting interaction with the 3' end initiation site of the RNA. This idea is interesting; however, the data presented is not sufficient to support the model. The minigenome WT and mutants bind the polymerase NS5 with low affinity (Figure 10 and subsection “Genome cyclization reduces the affinity of NS5 for vRNA by inducing the unwinding of the UFS”, first paragraph). However, RNA synthesis is efficient with the WT RNA (Figure 10), but not with mutant M1C (which is reconstituted). Also, it is uncertain why disrupted mutant M3A shows higher RNA synthesis than M3B? The presented data is over interpreted and the results do not support adequately the model proposed. Quantitative measurements should be provided and these discrepancies addressed.

5) Figure 8: the RNA-RNA complex formation with WT and mutated 5' 300 RNAs was analyzed in a range of concentration that are little informative in regard to differences in binding affinities. Kds must be calculated and presented.

6) The authors indicate that "UFS formation is essential for flavivirus replication" (subsection “Mechanistic model for the function of the UFS in flavivirus replication”, last paragraph). This RNA structure may function as enhancer of DENV4 replication but is not essential (Figure 5). In addition, disruption of the proposed UFS of DENV2 replicates similar to the WT virus, so the extrapolation to other flavivirus requires further studies.

---

## [Author Response]

Summary:

*Liu et al. propose that a previously undetected structural element in the 5' UTR of diverse flaviviruses is important for replication, but that upon genome cyclization this structure is disrupted by 5'-3' interactions. They propose that the RNA acts as a structural switch to coordinate cyclization and RdRP binding. Overall, the reviewers found that the model presented is reasonable but is not adequately supported by the experimental results. If the authors can provide additional data, analysis, and discussion to support this model, the manuscript may become suitable for publication in eLife.*

Essential revisions:

*1) The authors used many mutants, taking into account the impact on the linear form of the genome. However, the authors must also consider and discuss the impact of these mutations on RNA cyclization and on the internal loop structure (including UFS sequences) in the circular form. The predicted cyclization of DENV4 RNA is similar to that reported for JEV (Khromykh et al. 2003 JVI Figure 1), and different from other dengue serotypes. The model of genome cyclization of DENV4 RNA must be revised to include account possible hybridization of the U rich region of UFS with 3' end sequences (Figure 3, Figure 5 and Figure 7). For instance, in the M1 series, M1A may increase complementarity while M1B could reduce complementarity, similar analysis can be done for the M3 series. This possibility should be taken into account for the interpretation of results with all the mutants and the sequence/structure changes of the internal loop in the mutants should be analyzed.*

Thanks for the suggestions. We have also noticed that the U rich region of UFS is predicted to hybrid with the 3' end sequences by algorithms such as *mfold*, however, this hybridization pattern was not supported by our SHAPE data in this work (Figure 9). We have shown that when the 5'-300 nt RNA and 3' UTR RNA of DENV4 forms a complex, which is an in vitro mimic of circularized genome, the U-rich region exhibited high SHAPE reactivity, suggesting that this region becomes single-stranded after genome cyclization. Thus we believe that the U-rich region is not directly involved in interaction with the 3' UTR, at least in DENV4. Besides, we totally agree that the impact of mutations on RNA cyclization and on the internal loop structure in the circular form should also be discussed. To this end, using the DENV4 mini-genome sequence as the query sequence, the influence of M1 and M3 mutations on genome cyclization was assessed by using the *mfold* on line server (http://unafold.rna.albany.edu/?q=mfold/RNA-Folding-Form). The results of RNA structure prediction by *mfold* were provided as supplementary file, and the corresponding information has been added in the manuscript. It was shown that the overall genome cyclization pattern was only slightly affected by the M1 and M3 mutations ([Supplementary-material SD12-data]). The difference in free energy of the predicted circularized structures between WT and M1 mutants was only about 0.7 kcal/mol, whereas the M1A and M1B mutations decreased the local UFS structure by 3.4 and 2.3 kcal/mol, respectively, suggesting that these mutations mainly affected viral RNA replication through their disruptions of the UFS structure. Similar analysis was also performed for the M3 mutants ([Supplementary-material SD12-data]), indicating more markedly local structure changes over those of circularized structures. To further confirm the function of local UFS structure in vRNA replication, we designed a panel of mutations (Figure 3—figure supplement 1), which neither affect the pattern of RNA cyclization nor the internal loop structure in the circular form, and the corresponding mutants were assessed for replication efficiency. As shown in Figure 3—figure supplement 1, the disruption of UFS base pairing reduced vRNA replication greatly, whereas substituting of a few base pairs in the UFS only have moderate effects on vRNA replication. We believe that these new data and analysis would help to understand the function of UFS in vRNA replication in consideration of the impact of mutations on genome cyclization.

*2) The data shown in Figure 4 indicate that mutations within UFS alter SHAPE reactivity of SLA (which is the promoter for polymerase binding and activity), possibly explaining the reduced binding of the polymerase in the UFS mutants. Evaluation of the impact of UFS mutations on the overall molecule is crucial because the results regarding NS5 binding and RNA synthesis may be due to undesired changes within the promoter SLA. The authors must use the SHAPE data to generate predictions for these changes, for instance using RNAstructure software.*

Thanks for the reviewers’ insightful suggestions. We have used the *RNAstructure* software to generate results of RNA structure predictions by incorporating the constraints of SHAPE data as suggested, and the results of prediction indicated that the secondary structure of SLA was not affected by the DENV4 mutants M1A, M1C, M3A and M3C, thus the changes in NS5 binding and RNA synthesis of UFS mutants were not induced by unwanted disruption of the SLA element. The above results also confirmed that the UFS structure was disrupted/destabilized or reconstituted as designed. These results of prediction were provided as supplementary file ([Supplementary-material SD13-data]). In fact, we have speculated the observed changes of SHAPE reactivity of SLA and 5′ CS by UFS mutations M1A and M3A could be caused by the nature of data processing procedure of the QuShape software, which automatically determines the numbers of outliers in normalization procedure (described in QuShape Tutorial, which can be found at http://www.chem.unc.edu/rna/qushape/). As the M1A/M3A mutants exhibited more highly reactive sites (Figure 4), the automatically normalization process of QuShape would make the overall SHAPE reactivity of these mutants to become generally lower than the less-highly-reactive- sites-containing WT or “C” mutants.

*3) The authors find that NS5 affinity for the RNA is reduced when the UFS structure is disrupted by mutations. Considering that this observation is not caused by changes in SLA, the results do not agree with previous observations reported for DENV2. Specifically, mutations, deletions or complete replacement of UFS by unrelated sequences did not affect NS5 binding nor RNA synthesis* in vitro *(Filomatori et al. 2011 JBC Figure 2 and Figure 5, and Lodeiro et al. JVirol. 83,2, 993, 2009). Importantly, a substitution of the bottom half of UFS by As in the DENV2 infectious clone (mutant 6U/6A, Lodeiro et al. JV 2009) results in a replication competent virus similar to the WT. In this case UFS is completely disrupted and the virus shows efficient replication. These discrepancies suggest that the observations reported here are not applicable to other flaviviruses. The authors must better present their work in light of these potential differences; if the authors wish to conclude that their findings can be extended to other flaviviruses, additional data must to be provided to support this.*

Thanks for the reviewers’ suggestions. In order to extend our findings to the flavivirus genus, we have performed in vitro EMSA assay using JEV as the model, as well as mutagenesis analysis based on the infectious clone of Zika virus. The results of EMSA indicated that the UFS element is required for the efficient binding of JEV NS5Pol to viral 5' RNA (Figure 6—figure supplement 1),and we showed that disrupting of the UFS greatly reduced viral replication of Zika virus, whereas reconstitution of the UFS duplex restores Zika virus replication in BHK-21 cells (Figure 5—figure supplement 1). Together, these results demonstrated that the function of UFS element is indeed conserved in different flaviviruses. Regarding to the function of UFS in DENV2, although the biochemical assays mentioned by the reviewers indicated that UFS is not functional in DENV2 in vitro, the replication characteristics of the mutant 6U/6A (Lodeiro et al. 2009. J Virol. 83:993) was investigated only qualitatively. The original authors demonstrated that at 3 days post transfection, the DENV-positive cells were indistinguishable between WT and mutant 6U/6A, however, differences in replication were possible to be neglected as the authors did not provide the IFA results at 1-2 days post transfection, which is the period when most differences in flavivirus vRNA replication are observed. In fact, a similar mutant (Mutant poly (A)-10, Friebe et al. 2010. J Virol. 84:6103, Figure 7) was viable but showed reduced replication efficiency compared with its parental DENV2 replicon. Based on these reasons, we suggest that the previous evidences were not strong enough to conclude that the UFS is not functional in DENV2 vRNA replication. Even if the UFS was actually not a replication element in DENV2, that would make DENV2 to be an exception among flaviviruses based on our combined results. The relevant discussion was added into the first paragraph of the Discussion section.

*4) The authors propose that UFS formation is necessary for NS5 recruitment (linear form of the RNA) and when cyclization takes place, NS5 is released, promoting interaction with the 3' end initiation site of the RNA. This idea is interesting; however, the data presented is not sufficient to support the model. The minigenome WT and mutants bind the polymerase NS5 with low affinity (Figure 10 and subsection “Genome cyclization reduces the affinity of NS5 for vRNA by inducing the unwinding of the UFS”, first paragraph). However, RNA synthesis is efficient with the WT RNA (Figure 10), but not with mutant M1C (which is reconstituted). Also, it is uncertain why disrupted mutant M3A shows higher RNA synthesis than M3B? The presented data is over interpreted and the results do not support adequately the model proposed. Quantitative measurements should be provided and these discrepancies addressed.*

Thanks for the suggestions. We have analyzed the results of Figure 10 using ImageJ software, and demonstrated the RdRp activity as percentage of WT. In fact, the results in Figure 10 were not in conflict with the model which we have proposed. In Figure 10, we have shown that the mini-genomes are highly-circularized RNA molecules, and that means the UFS duplex is *not* present in these mini-genomes except for the CS-M mini-genome, which cannot circularize since the 5' CS element was mutated. Because of the above reasons, the mini-genomes except CS-M bind the polymerase NS5 with low affinity as shown in Figure 10. In the meantime, the presence of the core promoter, SLA, in these min-genomes and their pre-circularized nature should ensure the template activities of them in RdRp activity assay. Since the UFS duplex was not present in the mini-genomes, its function will not be reflected in the RdRp activity assay using mini-genomes as templates. The differences in RdRp activity between WT and M1C and between M3A and M3B should be caused by changes of primary sequence in these constructs, but not the alteration of UFS structure. In supporting of this idea, the M1C and M3C mutants were not able to reach the same level of replication as WT in replicon assay (Figure 3). Moreover, it has been reported (Friebe et al. 2010. J Virol. 84:6103) that substitution of U-rich sequence by adenosines is tolerated, but not by cytosines or guanosines. As the M1B and M1C mutants contains more guanosines in the U-rich region of the UFS than the other mutants, it is acceptable that these two mutants exhibited the lowest RdRp activity. We have added the above analysis into the corresponding Results section.

*5) Figure 8: the RNA-RNA complex formation with WT and mutated 5' 300 RNAs was analyzed in a range of concentration that are little informative in regard to differences in binding affinities. Kds must be calculated and presented.*

Thanks for the suggestions. We have re-performed the experiments shown in Figure 8 with a larger range of 5' RNA concentrations, and calculated the Kds, which are shown in Figure 8—figure supplement 2. The results indicated that the binding affinity of 5' M2C RNA was at least 10-fold higher than the other 5' RNA molecules, which have Kds ranging from approximately 70-250 nM.

*6) The authors indicate that "UFS formation is essential for flavivirus replication" (subsection “Mechanistic model for the function of the UFS in flavivirus replication”, last paragraph). This RNA structure may function as enhancer of DENV4 replication but is not essential (Figure 5). In addition, disruption of the proposed UFS of DENV2 replicates similar to the WT virus, so the extrapolation to other flavivirus requires further studies.*

Thanks for the suggestions. We have changed the corresponding description as suggested. In order to extend our findings to other flaviviruses, firstly, we performed in vitro EMSA assay using JEV as the model species (Figure 6—figure supplement 1), which demonstrating that the UFS element is also required for the efficient binding of JEV NS5Pol to the corresponding viral 5' RNA. Secondly, mutagenesis analysis targeting the UFS were performed based on the infectious clone of ZIKV (Figure 5—figure supplement 1). We showed that disrupting of the UFS greatly reduced viral replication, whereas reconstitution of the UFS duplex restores viral replication in BHK-21 cells. Together, these results suggested that the UFS plays a conserved role in vRNA replication of the flaviviruses.